# Melittin-lipid nanoparticles target to lymph nodes and elicit a systemic anti-tumor immune response

Xiang Yu[1,2,3], Yanfeng Dai [ID] [1,2,3], Yifan Zhao[1,2], Shuhong Qi[1,2], Lei Liu[1,2], Lisen Lu[1,2], Qingming Luo [ID] [1,2] & Zhihong Zhang [ID] [1,2 ✉]

Targeted delivery of a nanovaccine loaded with a tumor antigen and adjuvant to the lymph nodes (LNs) is an attractive approach for improving cancer immunotherapy outcomes. However, the application of this technique is restricted by the paucity of suitable tumor-associated antigens (TAAs) and the sophisticated technology required to identify tumor neoantigens. Here, we demonstrate that a self-assembling melittin-lipid nanoparticle (α-melittin-NP) that is not loaded with extra tumor antigens promotes whole tumor antigen release in situ and results in the activation of antigen-presenting cells (APCs) in LNs. Compared with free melittin, α-melittin-NPs markedly enhance LN accumulation and activation of APCs, leading to a 3.6-fold increase in antigen-specific CD8+ T cell responses. Furthermore, in a bilateral flank B16F10 tumor model, primary and distant tumor growth are significantly inhibited by α-melittin-NPs, with an inhibition rate of 95% and 92%, respectively. Thus, α-melittin-NPs induce a systemic anti-tumor response serving as an effective LN-targeted whole-cell nanovaccine.

[1] Britton Chance Center for Biomedical Photonics, Wuhan National Laboratory for Optoelectronics-Huazhong University of Science and Technology, 430074 Wuhan, Hubei, China. [2] MoE Key Laboratory for Biomedical Photonics, Collaborative Innovation Center for Biomedical Engineering, School of Engineering Sciences, Huazhong University of Science and Technology, 430074 Wuhan, Hubei, China. [3]These authors contributed equally: Xiang Yu, Yanfeng Dai. ✉email: czyzzh@mail.hust.edu.cn

Lymph nodes (LNs) are immunological organs that are strategically positioned at sites where antigens drain from peripheral tissues[1]. Once antigens are presented by antigen-presenting cells (APCs) residing in the tumor-draining LNs, an adaptive immune response is generated to defend against a tumor. Given the vital role of draining LNs in antitumor immunity, LNs have been considered a strategic target for cancer immunotherapy. For cancer vaccines, the targeted delivery of antigens and adjuvants to LNs to manipulate the LN micro-environment has a great promise[2]. Several recent studies have shown that LN-targeted delivery of nanovaccines awakened humoral and cellular immune responses to fight cancers[3,4]. However, the number of available tumor-associated antigens (TAAs), which are a key component of vaccines, to load onto the nanovaccines are very limited for most types of cancers[5,6]. Although several TAAs for melanoma have been defined, such as melanocyte differentiation antigens (MDAs), the immunogenicity of TAAs is highly variable among individuals, and TAAs can undergo immune-editing to escape immune recognition during tumor development[7]. In addition, neoantigens that arise as a consequence of tumor-specific mutations have been proved to be of particular relevance to tumor control[8–10], but the prediction of individualized neoantigens is mainly restrained by sophisticated technology[11,12].

Compared to above specific tumor antigens, whole-cell tumor antigens provide a broad spectrum of tumor antigens, thereby avoiding the costly and time-consuming process of identifying TAAs or neoantigens in a particular type of cancer[13]. Importantly, vaccines based on whole-cell tumor antigens can potentially elicit a stronger antitumor immune response, greatly decreasing the chance of tumor escape and recurrence. Based on these observations, we hypothesized that the ideal LN-targeted nanovaccines should utilize whole-cell tumor antigens rather than a single TAA, neoantigen or model antigen to generate a robust immune response against multiple antigen epitopes. Recently, Shi et al.[14] developed a novel chitosan nanoparticle (NP) that was loaded with whole tumor-cell lysates as the source of whole-cell tumor antigens to target resident dendritic cells (DCs) in LNs. However, the loaded whole-cell tumor antigens were generated from the repeated freezing and thawing of the tumor cells in vitro, which involved a substantial time commitment and inconveniences in clinical practice. Some approaches have been proposed to promote the release of whole-cell tumor antigens in situ, such as local radiotherapy, chemotherapy, and oncolytic viruses. Although these approaches can turn tumor into vaccines, the antitumor immune response is modest and must be combined with immuno-modulators (GM-CSF, CD40L, Flt3L, etc) to activate APCs[15,16]. Therefore, there is an urgent need to develop an effective LN-targeted whole-cell nanovaccine that can promote the release of whole-cell tumor antigens in situ and can activate APCs in LNs.

Melittin is the major component of European bee venom and has been used in traditional medicine to treat various diseases through its transdermal administration[17]. This peptide can induce tumor necrosis or apoptosis by disrupting cell membranes, accompanied with the release intracellular contents such as whole-tumor antigens and damage-associated molecular patterns[18]. Meanwhile, melittin, as a cationic host defense peptide, processes a wide variety of immunomodulatory effects[19,20]. However, these effects are moderate and insufficient to elicit a robust antitumor immune response because of the narrow safe dose range and hemolysis side effect of melittin[21]. In addition, the efficacy of melittin is also limited by unfavorable distribution because upon the subcutaneous (s.c) injection of small molecules (<16–20 kDa), these molecules preferentially enter blood circulation and are rapidly metabolized[22,23], preventing them from maximizing the immunomodulatory effects in the LNs.

Previously, we developed a high-density lipoprotein-mimicking peptide-phospholipid scaffold (named α-peptide-NP), whose structure could be precisely controlled by an α-helical peptide[24]. Then, we successfully loaded melittin onto the scaffold to form an ultrasmall (10–20 nm) melittin-lipid nanoparticle (named α-melittin-NP), and we confirmed that α-melittin-NPs efficiently shielded the positive charge of melittin within the phospholipid monolayer, resulting in reduced cytotoxicity to red blood cell (RBC)[25]. Therefore, we hypothesize that the intratumoral injection of α-melittin-NPs has two potential advantages for eliciting robust antitumor effects. First, α-melittin-NPs maintain the feature of melittin to directly induce tumor cell apoptosis or necrosis, thus releasing whole-tumor antigens in situ and bypassing the need of TAA loading on the nanovaccine. Second, α-melittin-NPs are the required size for an optimal LN-targeted nanovaccine that can efficiently drain into lymphatic capillaries and lymph node, thus providing best location for melittin to exert its full immuno-modulatory effect. In this study, we evaluate the ability of α-melittin-NPs to target the LN and reshape the immune micro-environment. These data show that α-melittin-NPs rapidly and efficiently drain to LNs and subsequently activate APCs, including macrophages and DCs. Meanwhile, fluorescence imaging data show that α-melittin-NPs induce the release of fluorescent model antigens in vivo. Furthermore, we verify the vaccine effect of α-melittin-NPs in a bilateral flank B16F10 tumor model. In situ vaccination with α-melittin-NPs successfully elicits systemic humoral and cellular immune responses, resulting in the elim-ination of 70% of primary tumors and 50% of distant tumors. Thus, α-melittin-NPs possess the ability to promote the release of whole-tumor antigens and the activation of the APCs and can serve as a promising LN-targeted whole-cell nanovaccine for cancer immunotherapy.

## Results

**α-melittin-NPs enhance the LN uptake of melittin.** The most critical factor that affects LN uptake is size, and the optimal size ranges from 10 to 100 nm[26,27]. Given that ultrasmall size (10–20 nm) is one of the attractive characteristics of the scaffold (α-peptide-NP) and α-melittin-NP, we assessed LN accumula-tion of melittin, α-peptide-NPs, and α-melittin-NPs labeled with fluorescein isothiocyanate (FITC) after s.c. injection. Wide-field fluorescence imaging data showed that the s.c. injection of α-melittin-NPs, as well as the α-peptide-NPs scaffold led to their substantial accumulation in inguinal LNs (ILNs) and axillary LNs (ALNs) (Fig. 1a), but not in other organs (Supplementary Fig. 1). However, FITC-melittin was not detected in ILNs and ALNs after s.c. injection (Fig. 1a). The inefficient uptake of melittin into the LNs is consistent with the small molecular weight of melittin (2840 Da), which can cause it to be directly absorbed into the blood from the injection site. Indeed, we found that approximately 33.6% red blood cells (RBCs) were FITC-positive (Fig. 1b, c), indicating the entry of melittin into the blood after s.c. injection. In addition, schistocytes were observed in peripheral blood smears in the melittin group (Supplementary Fig. 2), suggesting the occurrence of hemolysis, which is the main side effect of melittin. By further analyzing the cellular dis-tribution of α-melittin-NPs in LNs, we found that the percen-tages of macrophages and DCs that engulfed α-melittin-NPs were 38% and 10%, respectively, with little uptake observed in B cells and T cells in ILNs (Fig. 1d). Immunofluorescence staining also revealed α-melittin-NPs were mainly located in F4/80[+] macrophages and CD11c[+] DCs and were excluded from CD3[+] T and B220[+] B cells in ILNs and ALNs (Fig. 1e and Supplementary Fig. 3). To identify whether α-melittin-NPs can stimulate the activation of LN macrophages and DCs, the expression levels of

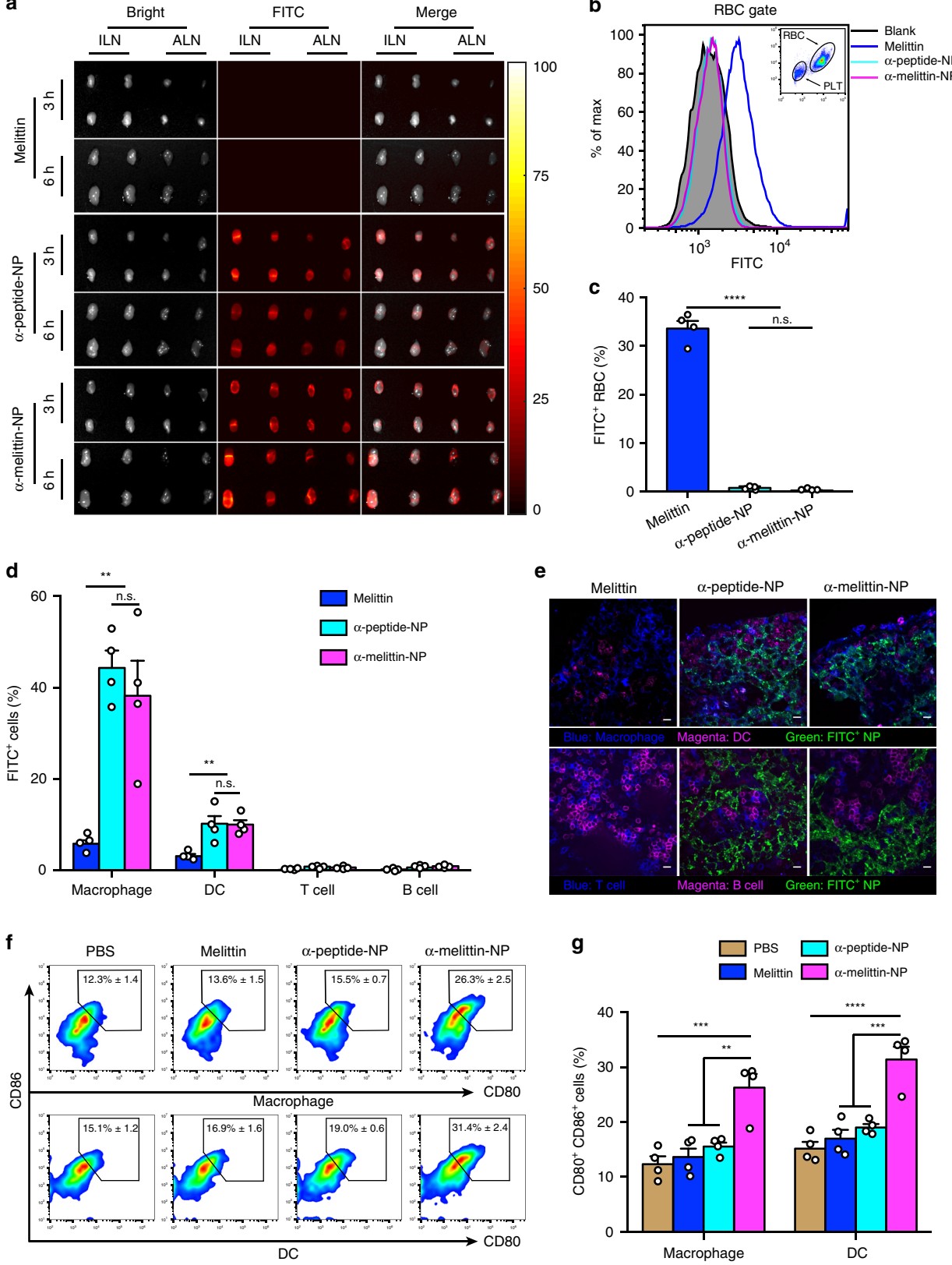

costimulatory molecules were detected using flow cytometry at 24 h after s.c. injection. Compared to phosphate-buffered saline (PBS), α-melittin-NPs induced the average percentage of CD80+ CD86+ macrophages and DCs in ILNs to increase from 12.3% to 26.3% and 15.1% to 31.4%, respectively (Fig. 1f, g). However, melittin and α-peptide-NPs had no obvious effects on expression of CD80 and CD86 on macrophages and DCs in ILNs (Fig. 1f, g) and ALNs (Supplementary Fig. 4). Taken together, these results suggest that the α-peptide-NP scaffold remarkably enhances the uptake of melittin in LNs, thus providing the precondition for the subsequent immunoactivation of macrophages and DCs in LNs by α-melittin-NPs.

**Fig. 1 α-melittin-NPs efficiently flow into the LN and stimulate the activation of APCs. a** Fluorescence images of excised LNs from C57BL/6 mice ($n = 4$ per group) subcutaneously injected with 20 nmol FITC-melittin, FITC-α-peptide-NPs, and FITC-α-melittin-NPs (quantification was based on the FITC content). ILN, inguinal lymph node. ALN, axillary lymph node. Time points: 3 h and 6 h. **b, c** Representative histograms (**b**) and percentages (**c**) of FITC+ RBC at 3 h after s.c. injection ($n = 4$ biologically independent samples). PLT, platelets. RBC, red blood cell. **d, e** Flow cytometry (**d**) and immunofluorescence analysis (**e**) of the uptake of FITC-melittin, FITC-α-peptide-NPs, and FITC-α-melittin-NPs by the immune cells in ILNs at 3 h after s.c. injection. Blue and magenta represent F4/80+ macrophages and CD11c+ DCs, respectively, in upper panel and represent CD3+ T cells and B220+ B cells, respectively, in lower panel. Green represents FITC-labeled melittin, α-peptide-NPs, and α-melittin-NPs. Representative images from three independent experiments are shown. Scale bar, 10 μm. **f, g** Representative flow cytometry plots (**f**) and quantitative data (**g**) for CD80 and CD86 expression on macrophages and DCs in ILNs ($n = 4$ per group) at 24 h after s.c. injection with 35 nmol melittin, α-peptide-NPs, and α-melittin-NPs (quantification was based on the peptide content). The histograms on the right show the percentage of activated APCs. Data are shown as the mean ± SEM. n.s. not significant, **$P < 0.01$, ***$P < 0.001$, and ****$P < 0.0001$, as analyzed by one-way ANOVA with Bonferroni's post hoc test (**c**, **d**, **g**). Source data are provided as a Source Data file.

**Cytotoxic effects of α-melittin-NPs on tumor cells and APCs.** To determine whether α-melittin-NPs are safe for APCs and are cytotoxic for tumor cells, different concentrations of α-melittin-NPs were incubated with bone marrow-derived dendritic cells (BMDCs), bone marrow-derived macrophages (BMDMs), and B16F10 melanoma cells. Free melittin and α-peptide-NPs were used as controls. Time-lapse fluorescent imaging revealed the appearance of propidium iodide (PI) signals and the disappearance of fluorescent protein signals in cells, which were used to dynamically visualize the cytotoxicity of α-melittin-NPs and melittin over the incubation time. As shown in Fig. 2a, b, α-melittin-NPs (10 μM) did not induce the obvious appearance of PI signals and the disappearance of fluorescent proteins in BMDCs and BMDMs after incubation for 1 h. In contrast, almost all BMDCs and BMDMs were stained by PI, and the intrinsic signals of fluorescent proteins disappeared after incubation with free melittin (5 μM) for 1 h. For B16F10 tumor cells, α-melittin-NPs seemed to maintain the killing effect of melittin (Fig. 2c). We then prolonged the incubation time to 24 h to further assess the cell viability at different concentrations by using (3-(4,5-dimethylthiazol-2-yl)-5-(3-carboxymethoxyphenyl)-2-(4-sulfophenyl)-2H-tetrazolium) (MTS) assays. Compared with the half-maximal inhibitory concentrations (IC50s) of free melittin (2.61 μM for BMDCs and 0.97 μM for BMDMs), those of α-melittin-NPs appeared to have decreased cytotoxicity to BMDCs and BMDMs, as indicated by significantly increased IC50 values (30.41 μM for BMDCs and 22.82 μM for BMDMs) (Supplementary Fig. 5). As negatively charged phosphatidylserine and O-glycosylated mucins are overexpressed in the plasma membrane of many cancer cells, thereby causing these membranes to carry a slightly higher net negative charge than those of normal eukaryotic cells[28], we speculated that the reason for the differential killing effects of α-melittin-NPs is probably related to membrane potential-mediated cellular binding. To further confirm this hypothesis, cell membrane potential was measured using a Malvern Zetasizer Nano-ZS90 instrument. The results showed that the zeta potential of B16F10 cell membrane (−27.65 ± 0.93 mV) was more negative than that of BMDCs (−14.64 ± 1.87 mV) and BMDMs (−10.78 ± 1.57 mV) (Supplementary Fig. 6). Next, we compared the ability of cellular binding of α-melittin-NPs to these three different cells using flow cytometry. The results revealed that the B16F10 cells captured a dramatically greater amount of FITC-α-melittin-NP than BMDMs at various concentrations during 1 h and 3 h incubation (Fig. 2d and Supplementary Fig. 7). We also noted that the there was no difference in the mean fluorescent intensity (MFI) of FITC between B16F10 cells and BMDCs during 1 h incubation (except 1.25 μM) (Supplementary Fig. 7), but as the incubation time was prolonged to 3 h and the concentration increased, B16F10 cells also displayed a significantly higher MFI value than BMDCs (Fig. 2d). To observe the cellular distribution of FITC-α-melittin-NPs in these three different cells in vitro, BMDCs and BMDMs were isolated from

mT/mG mice that express a strong red fluorescence protein (tdTomato) in the membrane systems (plasma membrane, lysosome, etc) of all cell types. Confocal imaging data showed that FITC-α-melittin-NPs displayed remarkably stronger fluorescent intensity in B16F10 cells than in BMDCs and BMDMs after incubation for 3 h in a 10 μM concentration (Fig. 2e).

More interestingly, FITC-α-melittin-NPs were mainly distributed in the cell membrane of B16F10 cells, but they were distributed in the intracellular membranes of BMDCs and BMDMs. This finding means that FITC-α-melittin-NPs quickly move through endocytosis to the intracellular membranes after binding. As the lytic activity of melittin is mainly associated with its ability to disturb cell membrane integrity by incorporating into phospholipid bilayers[29,30], the distribution of α-melittin-NPs in intracellular membranes makes APCs resistant to the plasma membrane permeabilization-dependent necrosis. In addition to zeta potential, cholesterol has been reported to have an inhibiting effect on the lytic activity of melittin to erythrocytes[31]. Next, we estimated the cholesterol content in the three different cells using an Amplex® Red reagent-based assay. However, the data showed that the cholesterol level in B16F10 cells was significantly higher than that in BMDCs and BMDMs (Supplementary Fig. 8). These results suggest that membrane potential-mediated cellular binding and subsequent spatial distribution are, at least in part, the main reasons for differential killing effects in a certain concentration range.

Further, we assessed the ability of α-melittin-NPs to induce the apoptosis and necrosis of tumor cells and the release of endogenous tumor antigens in vivo. tfRFP-B16F10 tumor cells that expressed tetrameric far-red fluorescent protein (tfRFP) were used to visualize the release of endogenous antigens[32]. The fluorescence imaging of tumor tissue sections showed that both melittin and α-melittin-NPs induced the disappearance of fluorescent model antigens in the tumor section after 24 h, indicating the changes in membrane penetrability and the release of the tumor antigens (Supplementary Fig. 9). The immunofluorescence imaging of TUNEL staining indicated that TUNEL-positive cells were also induced by melittin and α-melittin-NPs. It is noticeable, however, that TUNEL-positive cells also displayed the disappearance of fluorescent model antigens (plasma membrane permeabilization). Thus, α-melittin-NPs can shield the cytotoxicity of melittin to APCs, but maintain the ability of melittin to kill B16F10 tumor cells and induce the release of whole tumor-cell antigens in vivo.

**Enhanced therapeutic efficacy of α-melittin-NPs.** Given the confirmed immunomodulatory effect of α-melittin-NPs in LNs and the inherent property of melittin to induce the release of whole-tumor antigens, we evaluated the therapeutic efficacy of α-melittin-NPs as an LN-targeted whole-cell nanovaccine by establishing a bilateral flank tumor model. In the tumor model, the tumor on the left flank was used as a reservoir of tumor antigens, and the tumor on the other side was used to evaluate the effect of

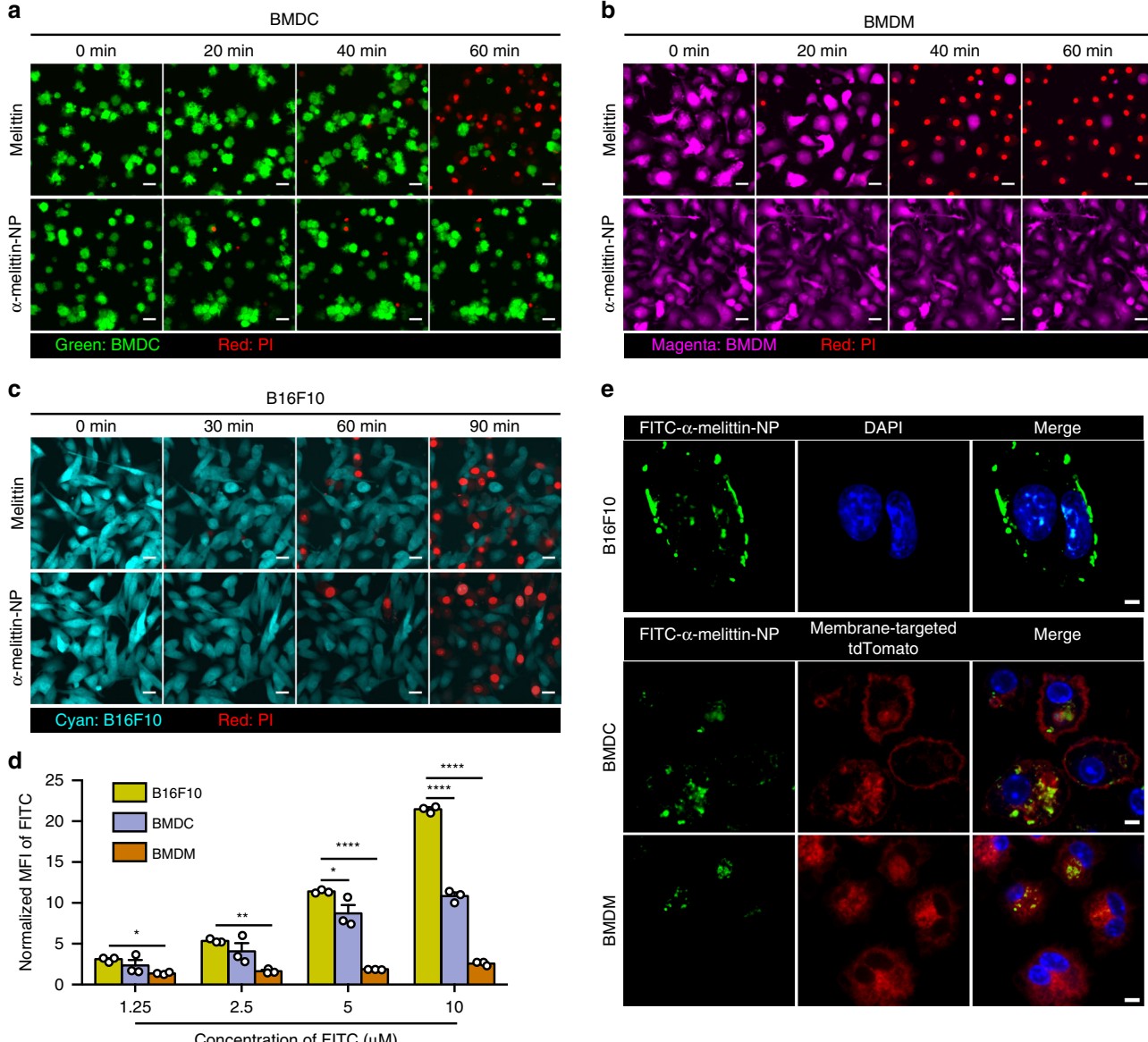

**Fig. 2 Cell viability of APCs and B16F10 cells after exposure to melittin and α-melittin-NPs. a–c** Real-time and dynamic imaging of BMDCs (**a**), BMDMs (**b**), and B16F10 cells (**c**) after incubations with free melittin (5 μM) and α-melittin-NP (10 μM). Green: BMDC, magenta: BMDM, cyan: B16F10. Red indicates PI. BMDCs and BMDMs were isolated from Actb-EGFP C57BL/6 mice in which EGFP is expressed uniformly in all cells except the erythrocytes and hair. Representative images from three independent experiments are shown. All scale bars represent 10 μm. **d** Evaluation of cellular-binding ability by analyzing the MFI of FITC-α-melittin-NPs in B16F10 cells, BMDCs and BMDMs ($n = 3$ per group). Incubation time: 3 h. MFI: mean fluorescent intensity. The MFI values were normalized according to minimum in each type of cell. **e** Representative immunofluorescence imaging of cellular binding of FITC-α-melittin-NPs (10 μM) to B16F10 cells, BMDCs and BMDMs. BMDCs and BMDMs were isolated from mT/mG mice that express a strong red fluorescence protein (tdTomato) in the membrane systems (plasma membrane, lysosome, etc) of all cell types. Incubation time: 3 h. Blue: DAPI, green: FITC-α-melittin-NPs, red: membrane-targeted tdTomato. Representative images from three independent experiments are shown. Scale bar: 5 μm. Data are shown as the mean ± SEM. *$P < 0.05$, **$P < 0.01$ and ****$P < 0.0001$, as analyzed by one-way ANOVA with Bonferroni's post hoc test (**d**). Source data are provided as a Source Data file.

vaccine. Specifically, we implanted B16F10 cells into the left and right flanks of mice on days 0 and 4, respectively, and the mice were treated with intratumoral injections of 35 nmol melittin, α-peptide-NPs, and α-melittin-NPs in PBS, with a total volume of 50 μl, on days 7 and 9 (Fig. 3a). The tumor growth curve showed that α-melittin-NPs dramatically suppressed the tumor growth on the left flank (injected tumor) and right flank (distant tumor). At 20 days post left tumor inoculation, compared with the PBS group, mice treated with α-melittin-NPs showed a 95% decrease in the injected tumor size and a 92% decrease in the distant tumor size (Fig. 3b–d and Supplementary Fig. 10). Attractively, while

continually monitoring the tumor growth for 60 days, we found that the percent of tumor rejection was 70% in the left flank and 50% in the right flank (Fig. 3e, f). In addition, compared to the α-melittin-NP group, the melittin group showed a less dramatic decrease in tumor size (37% in the left flank and 66% in the right flank) and a lower percentage of tumor rejection (10% in the left flank and 0% in the right flank) (Fig. 3e, f). To exclude the possible direct killing effect of α-melittin-NPs on the contralateral tumor cells, we detected the distribution of FITC-labeled α-melittin-NPs after intratumoral administration into the tumor in the left flank. The quantitative analysis of the FITC localization showed that

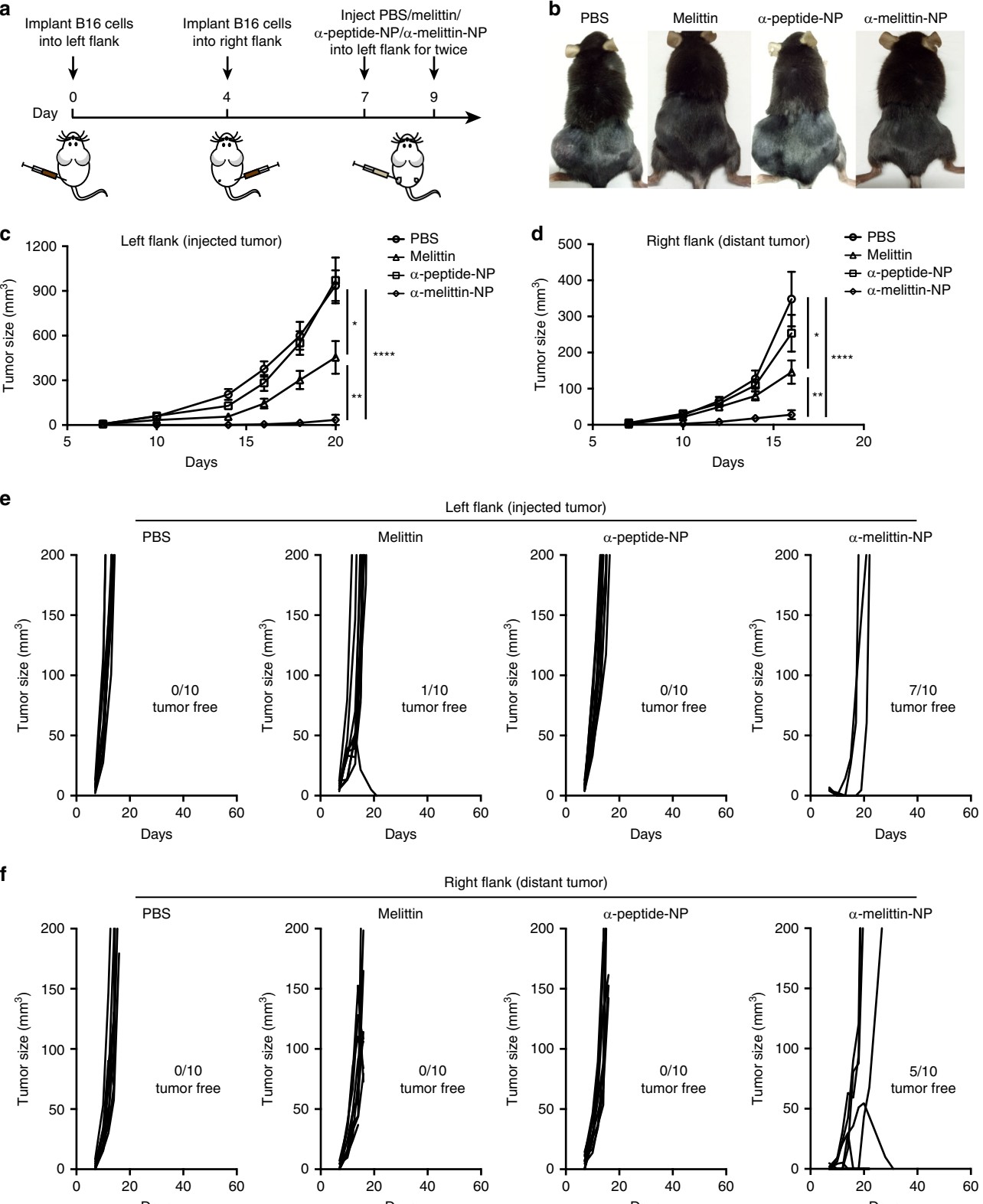

**Fig. 3 α-melittin-NPs as an in situ vaccine delay tumor growth and mediate systemic tumor control. a** Treatment scheme. B16F10 cells were implanted into the left and right flanks of mice ($n = 10$ per group) on days 0 and 4, respectively. The mice were treated with intratumoral injections of 35 nmol melittin, α-peptide-NPs, and α-melittin-NPs (quantification was based on the peptide content) on days 7 and 9. **b** Representative pictures of mice treated with the scheme described above. All individuals are shown in Supplementary Fig. 10. **c**, **d** Tumor growth of the injected tumor (**c**) and distant tumor (**d**). **e**, **f** Individual tumor growth kinetics of the injected (**e**) and distant tumor (**f**). Data are shown as the mean ± SEM. *$P < 0.05$, **$P < 0.01$, and ****$P < 0.0001$, as analyzed by one-way ANOVA with Bonferroni's post hoc test (**c**, **d**). Source data are provided as a Source Data file.

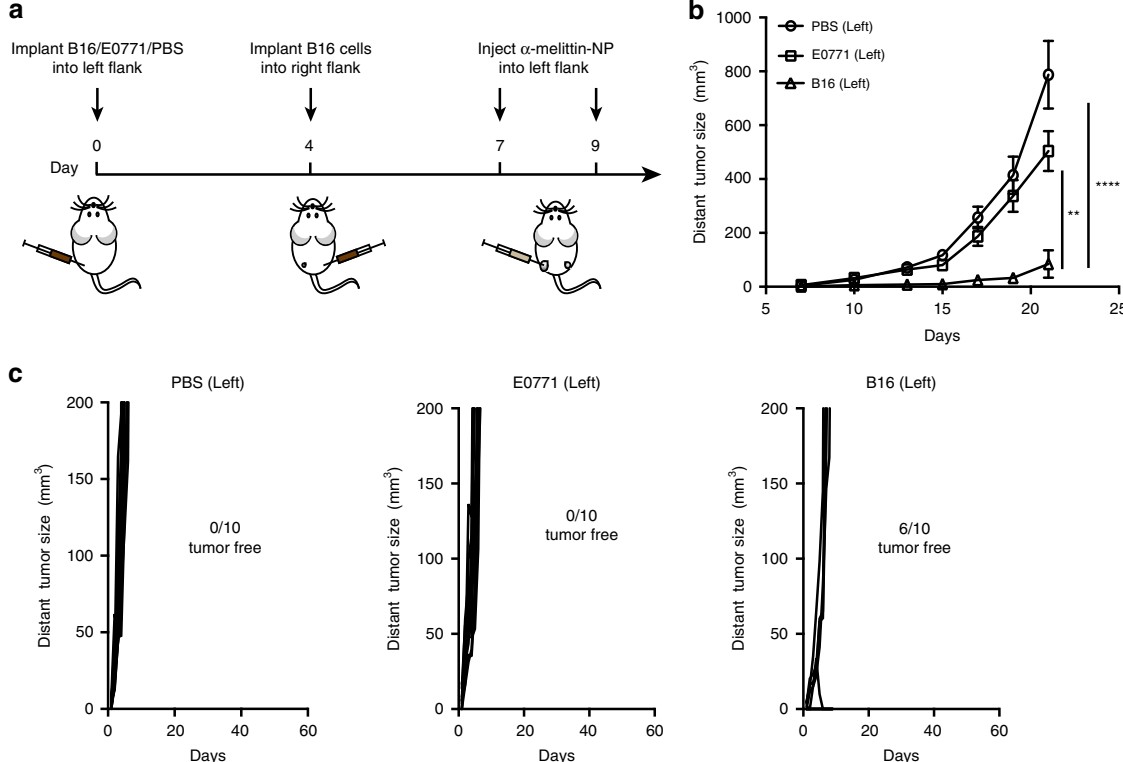

**Fig. 4 Systemic antitumor effect is specific for whole-cell tumor antigens. a** Treatment scheme. Mice ($n = 10$ per group) were inoculated in the left flank with B16F10 cells, E0771 cells, or PBS on day 0. After 4 days, B16F10 cells were implanted into the right flank. The mice were treated with intratumoral injections of 35 nmol α-melittin-NPs (quantification was based on the peptide content) in PBS, with a total volume of 50 μl, on days 7 and 9. **b** Growth of tumors in the right flank. **c** Individual tumor growth kinetics of the right tumors. Data are shown as the mean ± SEM. **$P < 0.01$ and ****$P < 0.0001$, as analyzed by one-way ANOVA with Bonferroni's post hoc test (**b**). Source data are provided as a Source Data file.

α-melittin-NPs were restricted mainly to the injected tumor site and the draining LNs and were nearly undetectable in the contralateral tumor, its draining LNs and other organs (e.g., heart, liver, spleen, lung, kidney, and brain) (Supplementary Fig. 11). These results indicate that the reduction in the growth of distant tumors was due to the systemic immune activation elicited by the intratumoral injection of α-melittin-NPs rather than due to a direct killing effect of α-melittin-NPs on distant tumor cells.

In addition, to determine whether the vaccine effect of α-melittin-NP was durable, we screened the cured mice and implanted the same tumor cells into the right flank at 65 days after the first inoculation. We found that α-melittin-NP treatment caused prolonged survival, and 20% of the mice completely rejected rechallenged cells (Supplementary Fig. 12). Taken together, these results suggest that α-melittin-NPs can be used as an LN-targeted whole-cell nanovaccine to delay tumor growth and mediate durable systemic antitumor responses.

**α-melittin-NPs elicit specific and potent antitumor immunity.**
To examine whether the observed therapeutic effect in the right flank (distant tumor) was an antigen-specific immune response, we implanted different types of tumor cells (B16F10 and E0771) in the left flank, and then implanted same B16F10 tumor cells in the right flank to monitor their growth (Fig. 4a). We found that compared to E0771 tumor cell, the administration of α-melittin-NPs into the B16F10 tumor in the left flank resulted in an 83% decrease in right tumor size, and the percent of right tumor rejection reached 60% (Fig. 4b, c). However, the administration of α-melittin-NPs into the E0771 tumor in the left flank had no significant inhibitory effect on the growth of right B16F10 tumor

and failed to induce the regression of right B16F10 tumors (Fig. 4b, c). Thus, the systemic antitumor response induced by α-melittin-NPs is restricted by the type of antigen reservoir.

After observing the therapeutic efficacy of α-melittin-NPs, we next assessed tumor antigen-specific cellular and humoral immune response. Given the critical role of LNs in the antitumor response and the immunomodulatory effect of α-melittin-NPs in LNs, lymphocytes were isolated from the tumor-draining LNs of mice that were treated with the scheme shown in Fig. 3a, and were restimulated with DCs pulsed with B16F10 tumor lysates. We used flow cytometry to analyze the cytokine expression in T cells. The data showed that compared with PBS group, α-melittin-NPs induced increases in the frequencies of IFN-γ+ CD8+ (12.2-fold) and IFN-γ+CD4+ (7.2-fold) T cells at day 21 after tumor implantations (Fig. 5a–c) but no differences emerged at 14 days (Supplementary Fig. 13a, b). We also observed that melittin induced moderate increases in the frequencies of IFN-γ+ CD8+ T cells (3.3-fold) compared to PBS, providing a partial explanation for the inhibitory effect of melittin on the size of the distant tumor.

In parallel, we also collected serum to evaluate the endogenous antibody response on 21 days after tumor inoculation. The B16F10 tumor cells were incubated with 5% serum for 0.5 h and were then stained with a Dylight649-conjugated mouse IgG-specific secondary antibody. The flow cytometry data showed that the percent of IgG+ cells in the melittin and α-melittin-NP groups was ~2-fold higher than those in the PBS and α-peptide-NP groups at day 21 (Fig. 5d, e) but no differences emerged at 14 days (Supplementary Fig. 13c). It is important to note, however, that there was no difference in the percent of IgG+ cells between the melittin and α-melittin-NP groups at days 21,

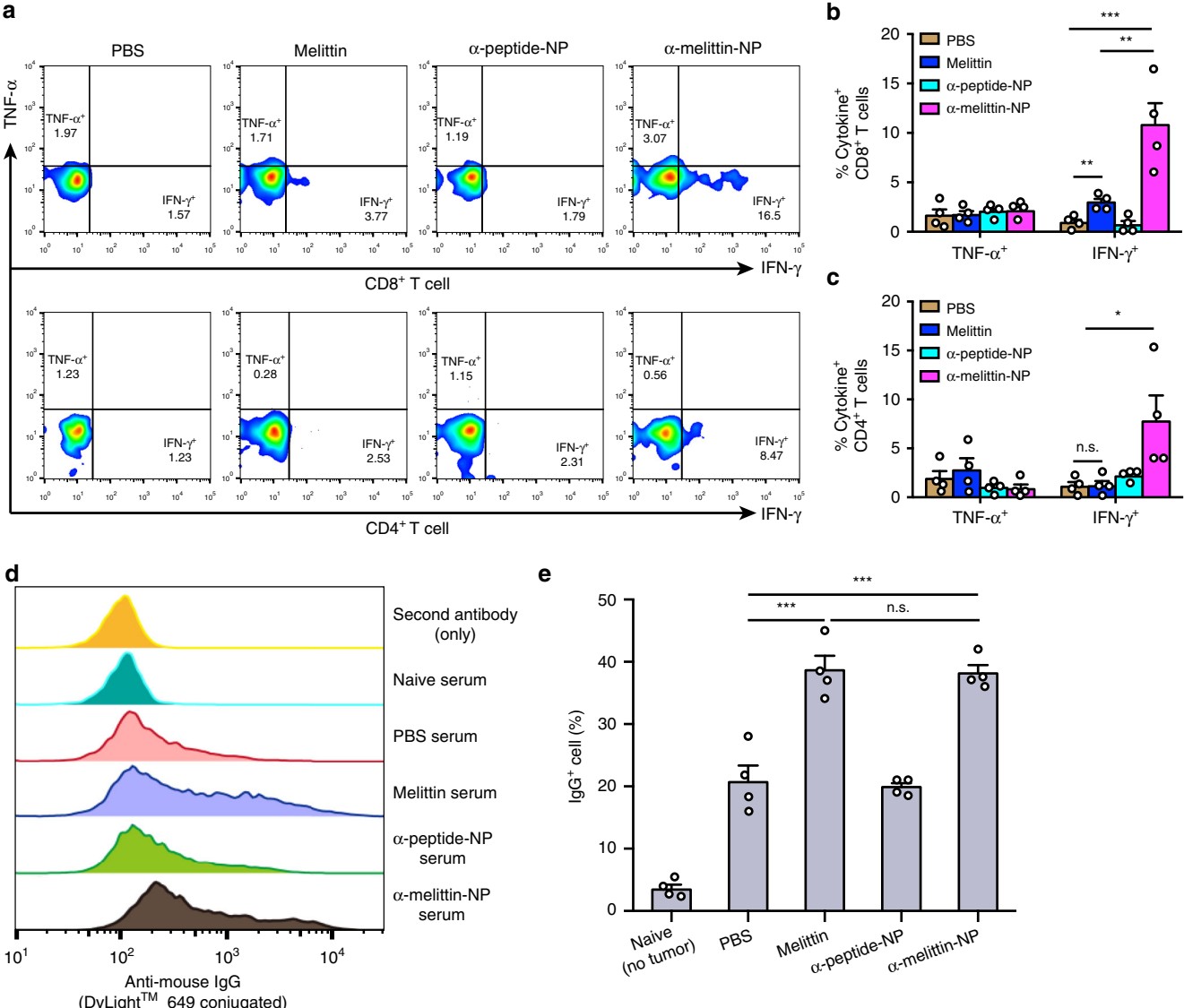

**Fig. 5 α-melittin-NPs induce antigen-specific T cells and antibody responses. a–c** C57BL/6 mice ($n = 4$ per group) were treated as described above (Fig. 3a), on day 21, the lymphocytes isolated from the tumor-draining LNs were restimulated with DCs pulsed with B16F10 tumor lysates and were analyzed by flow cytometry with intracellular cytokine staining. Representative flow cytometry plots (**a**) and cumulative results (**b, c**) are shown. **d, e** The B16F10 tumor cells were incubated with 5% serum that was collected from treated mice and age-matched naïve mice ($n = 4$ per group). Subsequently, these cells were stained with a DyLight649-conjugated mouse IgG-specific secondary antibody and were analyzed by flow cytometry. Flow cytometry plots (**d**) and statistical percentages (**e**) are shown. Data are shown as the mean ± SEM. n.s. not significant, *$P < 0.05$, **$P < 0.01$ and ***$P < 0.001$, as analyzed by one-way ANOVA with Bonferroni's post hoc test (**b, c, e**). Source data are provided as a Source Data file.

suggesting that the enhanced vaccine efficacy of α-melittin-NPs was mainly due to cellular immunity rather than antibody response. Taken together, these results indicate that α-melittin-NPs can induce both cellular and humoral immune responses to whole-cell tumor antigens.

**Features of immune microenvironment in the distant tumor.** The infiltration of leukocytes in the distant tumors are also factors that are vital for an effective vaccine response. Thus, we further analyzed lymphocyte infiltration in distant tumors at 14 and 21 days after left tumor implantation (Fig. 6a), and the gating strategy was shown in Supplementary Fig. 14. The flow cytometry data showed that, compared to the PBS, α-melittin-NP induced an increase in the numbers of innate immune components, including natural killer (NK) cells, monocytes and neutrophils, but not the adaptive components, including CD4+ and CD8+

T cells at 14 days after left tumor implantation (Fig. 6b, c). However, at 21 days, α-melittin-NP group exhibited a significant increase in the number of CD4+ (4.4-fold, $P = 0.0074$) and CD8+ T cells (3.7-fold, $P = 0.0243$) in addition to NK cells and monocytes (Fig. 6b, c). Immunofluorescent analysis of the distant tumors also revealed that CD4+ and CD8+ T cells were present at high density after α-melittin-NP treatment (Fig. 6d). Meanwhile, we also found that compared to PBS, melittin also could induce a moderate increase in the number of monocytes (3.7-fold, $P = 0.0125$) and CD4+ T cells (2.5-fold, $P = 0.0099$), but there was no change in CD8+ T cells. To understand the molecular mechanisms that drove increased leukocyte infiltration, we analyzed the intratumoral expression of cytokines/chemokines involved in leukocyte trafficking. The data showed that the intratumoral injection of α-melittin-NPs resulted in elevated levels of chemokines involved in T and NK cells recruitment (CCL3-4, CXCL10,

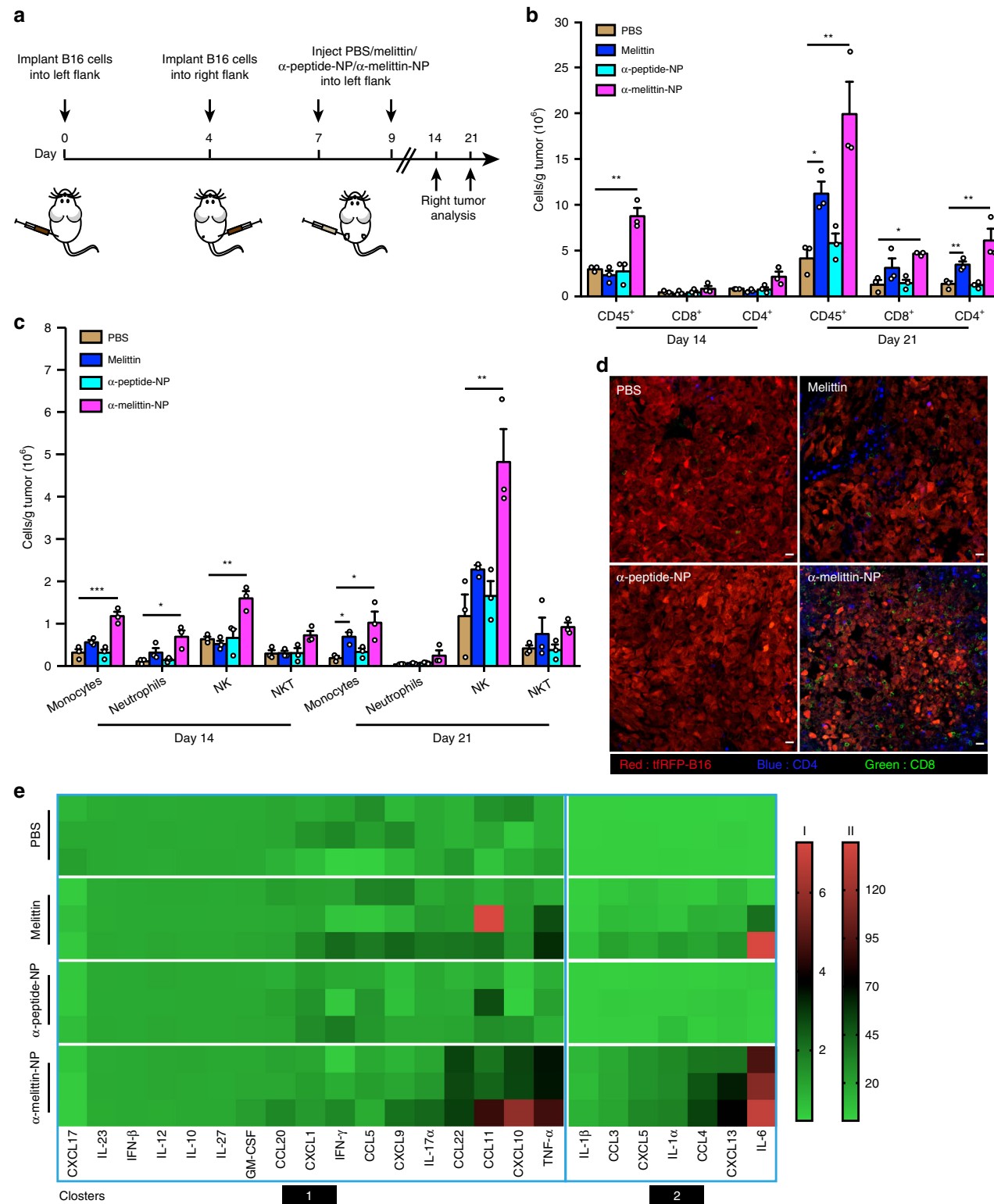

**Fig. 6 α-melittin-NPs induce the lymphocyte infiltration and dramatic changes in the cytokines/chemokines milieu in distant tumor. a** Treatment scheme. **b, c** Absolute numbers of adaptive immune cells (**b**) and innate immune cells (**c**) in the distant tumor ($n = 3$ per group per time point) were calculated from flow cytometry at 14 and 21 days after left tumor implantation. **d** Immunofluorescence images from distant tumors 21 days after left tumor implantation. Red: tfRFP-B16F10, blue: CD4+ T cell, green: CD8+ T cell. Representative images from three independent experiments are shown. Scale bar, 20 μm. **e** The levels of cytokines/chemokines in the tumor environment ($n = 3$ per group). According to the fold change relative to the untreated group, the expression levels of cytokines and chemokines were divided into two clusters. Clusters 1 and 2 share bars I and II, respectively. Data are shown as the mean ± SEM. n.s. not significant, *$P < 0.05$, **$P < 0.01$ and ***$P < 0.001$, as analyzed by one-way ANOVA with Bonferroni's post hoc (**b, c**). Source data are provided as a Source Data file.

and CCL22) and monocytes recruitment (CXCL5) (Fig. 6e). Meanwhile, the levels of proinflammatory cytokines (tumor necrosis factor (TNF)-α, interleukin (IL)-β, IL-1α, and IL-6) were also increased, indicating the formation of a beneficial inflamed tumor microenvironment. Thus, α-melittin-NP treatment induces significant remodeling of the distant tumor microenvironment, including immune cell infiltration and dramatic changes in the cytokine/chemokine milieu.

## Discussion

LN-targeted vaccines can make full use of resident APCs in the LN to present antigen and stimulate T cells to defend against tumor. Unlike conventional DC vaccines, LN-targeted vaccines do not require the manipulating of the APCs in vitro and, theoretically, have a broader application potential. However, the development of LN-targeted nanovaccines has been hindered by the paucity of TAAs and the complex technical constraints of identifying tumor neoantigens. Compared to TAAs or neoantigens, whole-tumor antigens can provide multiple antigenic epitopes to generate a broad polyclonal tumor-specific response. In this study, we demonstrated that α-melittin-NPs that were not loaded extra tumor antigens elicit a potent and long-lasting antitumor immune response by a one-stone-two-birds strategy. In respect to the mechanism, α-melittin-NPs affected both tumor cells and immune cells. On the one hand, α-melittin-NPs retained the characteristics of melittin that promoted the release of whole-tumor antigens in situ, and we also confirmed that the variety of immunogenic epitopes provided by whole-tumor antigens could activate both CD4$^+$ and CD8$^+$ tumor-specific T cells. On the other hand, with the help of the α-peptide-NP scaffold, α-melittin-NPs could efficiently drain to LNs and subsequently activated macrophages and DCs and also induced the infiltration of innate immune cells to synergistically control distant tumor growth, especially NK cells. With respect to the antitumor effects, α-melittin-NPs inhibited both primary and distant tumor growth. On the one hand, mice treated with α-melittin-NPs showed a 95% decrease in primary tumor size, and the percent of tumor rejection was 70%. Additionally, α-melittin-NPs induced a 92% decrease in distant tumor size, and the percent of tumor rejection was 50%. The mechanism described above for the in situ vaccine effect of α-melittin-NPs is shown in detail in Fig. 7.

As a natural host defense peptide (HDP)[33], melittin also processes a wide variety of immunomodulatory effects. For example, Nam et al.[34] showed that the administration of melittin along with a conventional vaccine (HBsAg/alum) induces the Th1 lineage development of CD4$^+$ T cells by increasing the expression of a Th1-specific cytokine in response to hepatitis B virus infection. Palm et al.[35] also demonstrated that in addition to phospholipase A2, bee venom and melittin induced antigen-specific IgG1 and IgE responses to admixed ovalbumin (OVA) after s.c. immunization. These immunomodulatory effects are relatively weak because of the hemolysis side effect and unfavorable distribution. In our study, we found that in a certain concentration range, α-melittin-NPs had a strong killing effect on tumor cells, but had little toxicity in BMDC and BMDM, thus providing a precondition for the subsequent immunomodulatory effects of α-melittin-NPs on APCs. Although the lytic activity of melittin is mainly associated with its ability to disturb cell membrane integrity by incorporating into phospholipid bilayers, nanoparticle-delivered melittin might elicit apoptosis after trafficking to intracellular membranes via activation of the intrinsic pathway[36]. Therefore, the different spatial distribution of α-melittin-NPs in B16F10 cells and APCs (BMDCs and BMDMs) seem to not explain simply the differential killing effects. A plausible possibility is the existence of a mechanism of anti-apoptosis. After all, necrosis is mainly an irreversible event, but apoptosis can be regulated. We also previously found that Birc5 (also known as survivin), a member of the inhibitor of apoptosis (IAP) gene family encoding negative regulatory proteins that prevent apoptotic cell death, was upregulated in liver sinusoidal endothelial cells (LSECs, another APC in liver) after the administration of α-melittin-NPs[37]. However, it is worth noting that α-melittin-NPs also exhibited cytotoxicity in APCs with the increase of melittin concentration. In addition, we found that α-melittin-NPs could induce the maturation of macrophages and DCs in LNs and cause dramatic changes in the cytokine/chemokine milieu in the tumor. These results were consistent with our recently published study about the immunomodulatory effects of α-melittin-NPs on LSECs[37]. We also noticed that the cytokine/chemokine secretion profiles were not uniform, probably because of the existence of strong intrinsic immune suppression environment in liver.

Small molecules or moderately sized macromolecules (<16–20 kDa or 10 nm) are primarily absorbed via blood capillaries after s.c. injection, whereas macromolecules (20–30 kDa or 10–100 nm) mainly enter lymphatic vessels[22]. As the molecular weight of melittin is 2840 Da, the s.c. injection of melittin may cause melittin to preferentially enter the blood rather than LNs, and our results also confirmed this hypothesis. We found that FITC-labeled melittin was not detected in ILNs and ALNs, but 33.6% of RBCs were FITC-positive. In addition, we observed the presence of schistocytes in peripheral blood smears in the melittin group, suggesting that melittin enters the bloodstream and causes hemolysis. When melittin was incorporated into the α-peptide-NP scaffold (10– 20 nm), α-melittin-NPs successfully drained to the LNs, where they were engulfed by macrophages (38%) and DCs (10%). However, the in vitro experiments showed that BMDCs captured a greater amount of α-melittin-NP than BMDMs. In fact, sinus macrophages are the first cells encountering and phagocytosing nanoparticle carried by the afferent lymphatics in vivo[38].

As intratumoral injection allows much higher concentrations of the immunostimulatory products in the tumor microenvironment than systemic administration, intratumoral treatment is still popular[39–42]. However, any intratumoral therapy requires access to the tumor site. The accessibility of primary melanomas and subcutaneous breast tumors provide interesting examples. For deep-seated malignant tumors, this strategy must turn to the guidance with B-ultrasonography or CT. For the evaluation of therapeutic vaccination, we chose a bilateral flank melanoma model that utilized the staggered implantation of tumors. In addition to the expected therapeutic effects of α-melittin-NPs, we also found that melittin could induce a certain degree of tumor inhibition; this result is not surprising because melittin can kill tumor cells to release tumor antigens. However, the incomplete response occurred in both α-melittin-NPs and melittin groups. Tumor progression involves the co-evolution of neoplastic cells together with tumor microenvironment, and heterologous cell types within tumors can actively influence the therapeutic response and shape resistance even, in cases in which immune cells actively drive the initial response to targeted therapies[43]. In some individuals, the CD8$^+$ T cells infiltrate may be concomitant with the elevated level of T-cell inhibitory receptors, such as T lymphocyte-associated antigen-4 (CTLA-4) and programmed death 1 (PD1), leading to the emergence of T-cell exhaustion. For example, while oncolytic virotherapy induced the infiltration of activated lymphocytes in tumors, the antitumor effect was unable to lead complete tumor regression because of treatment-induced adaptive immune resistance manifested by upregulation of CTLA-4 or PD1[44–46]. Therefore, we hypothesize that incomplete response induced by α-melittin-NPs in certain individual can be improved by combination strategies using checkpoint inhibitors.

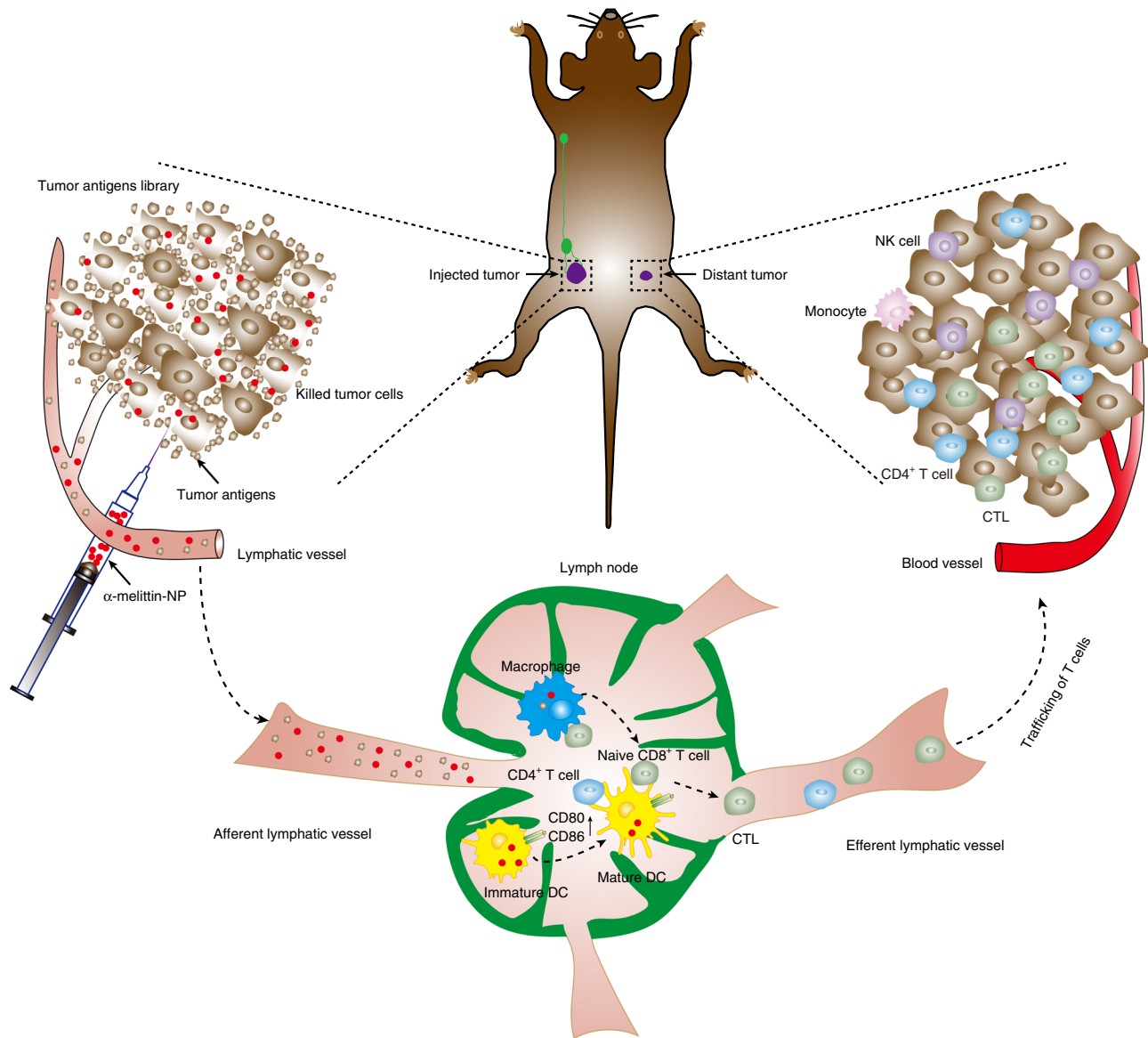

**Fig. 7 Schematic description of the mechanism of the in situ vaccine effect induced by α-melittin-NPs.** On the one hand, tumor cells are sensitive to α-melittin-NPs, and α-melittin-NPs maintain the ability of melittin to kill tumor cells and to promote the release of whole tumor-cell antigens in situ. On the other hand, α-melittin-NPs with reasonable size can successfully drain into LNs and activate macrophages and DC after s.c. injection. After the priming and activation of the effector T-cell response against whole-tumor antigens in LN, the activated effector T cells traffic through the circulation into the distant tumor bed and kill their target tumor cells. Meanwhile, α-melittin-NPs also induce the infiltration of innate immune cells, especially NK cells and monocytes.

Current practices for LN-targeted whole-cell nanovaccine involve in the preparation of tumor tissues by surgery as well as antigens loading in vitro. These steps require a substantial time commitment and may cause potential contamination. In our study, intratumoral direct injection of α-melittin-NPs turn the tumor into a vaccine factory without any manual manipulations in vitro. In addition, the α-melittin-NPs components contain phospholipid, cholesterol oleate and peptide. Both phospholipid and cholesterol oleate have perfect biocompatibility and are relatively inexpensive. The last few years have also seen a remarkable decrease in the cost of peptide synthesis due to cost of raw materials and technical improvements in peptide reverse phase flash chromatography[47]. More importantly, the application of nanotechnology and intratumoral injection greatly decrease the required dosage of melittin peptide, generating further cost reductions.

Altogether, we found that α-melittin-NPs that were not loaded with specific tumor antigen and adjuvant could serve as an excellent nanovaccine. On the one hand, α-melittin-NPs maintain the killing effect of melittin towards tumor cells and can induce the release of whole-tumor antigens in situ. On the other hand, α-melittin-NPs with optimal size can efficiently flow into the LNs and activate resident APCs. By using the bilateral flank B16F10 tumor model, our results indicated that α-melittin-NPs turn tumors into vaccines and induce a systemic immune response, resulting in delayed tumor growth and even in the complete regression of the distant tumor. α-melittin-NPs have strengths, including a simple preparation, good stability, and no side effects. These properties, combined with their inherent oncolytic properties and immunoregulatory effects, make α-melittin-NPs an ideal LN-targeted whole-cell nanovaccine with transformational clinical potential.

## Methods

**Materials**. Cholesterol oleate (CO), heparin, fluorescein isothiocyanate (FITC), and protease inhibitor cocktail were purchased from Sigma-Aldrich Co. (St. Louis, MO, USA). 1,2-dimyristoyl-sn-glycero-3-phosphocholine (DMPC) was obtained from Avanti Polar Lipids Inc. (Alabaster, AL, USA). The synthesis of α-peptide (DWFKAFYDKVAEKFKEAF-NH$_2$) and α-melittin (DWFKA-FYDKVAEKFKEAF-GSG-GIGAVLKVLTTGLPALISW-IKRKRQQ-NH$_2$) were entrusted to Apeptide Co., Ltd. (Shanghai, China).

**Mice and cells**. Wild-type C57BL/6 mice (Female, 6–8-week-old) were purchased from Hunan SJA Laboratory Animal Co., Ltd (Changsha, Hunan, China). Actb-EGFP C57BL/6 mice were kindly provided by Dr. Zhiying He (Second Military Medical University, Shanghai, China). mT/mG transgenic mice were obtained from the Jackson Laboratory (Bar Harbor, ME, USA). All animal studies were conducted in compliance with protocols that had been approved by the Hubei Provincial Animal Care and Use Committee and in compliance with the experimental guidelines of the Animal Experimentation Ethics Committee of Huazhong University of Science and Technology. The B16F10 cell line was purchased from the BOSTER Company (Wuhan, China). The E0771 cell line was kindly provided by Rong Xiang (Medical School of Nankai University, Tianjin, China) These cell lines were authenticated using short tandem repeat (STR) profiling and were mycoplasma-negative as determined by screening using the MycoProbe Mycoplasma Detection Kit (R and D Systems, Minneapolis, MN). B16F10 and E0771 cells were cultured in RPMI-1640 containing 10% fetal bovine serum (FBS) and 100 U/ml penicillin-streptomycin under 5% CO$_2$ at 37 °C in an incubator (Thermo, USA).

**Synthesis of nanoparticle**. A mixture of DMPC (3 μmol) and CO (0.2 μmol) in chloroform (300 μl, AR grade) was dried under nitrogen to form a uniform lipid film. Then, 1 ml PBS was added to the dried film, and the mixture was vortexed for at least 5 min to redissolve the adsorbed product. Subsequently, the mixture was sonicated for about 1 h at 48 °C until the solution is as clear as possible. α-melittin (0.19 μmol, hybridized peptide) and α-peptide (0.87 μmol, an ApoA1-mimetic peptide) were dissolved in 1 ml PBS and double distilled water, respectively. These two peptides were added to the lipid emulsion, and then stored overnight at 4 °C. During the incubation time, the interaction between amphipathic α-helix in peptide (α-melittin and α-peptide) and the lipid monolayer resulted in nanoparticles that appeared spherical in shape and possessed a small particle size. Meanwhile, the concentration of peptide increase means the reduction in the spherical size, and a ratio of 0.5:1 is the optimal weight ratio of peptide to lipid. After being concentrated by centrifugal filter units (30 Kd, Millipore, USA), the nanoparticles were purified using a fast protein liquid chromatography system with a HiLoad 16/60 Superdex 200 pg column (General Electric Healthcare, NY, USA). Particles eluted at a retention time of about 60 min (absorption peak at 280 nm) were collected and concentrated to 0.5 ml. To prepare the nanoparticles that were labeled with FITC, FITC was conjugated to the primary amines the peptides after the α-melittin-NPs and α-peptide-NPs had been prepared. The peptide concentration was measured using a CBQCA protein quantitation kit (Invitrogen Corporation, CA, USA). The zeta potentials of the α-melittin-NPs were measured using dynamic light scattering photon correlation spectroscopy on a Zetasizer Nano-ZS90 system (Malvern Instruments, Worcestershire, UK).

**Isolation of mouse BMDCs and BMDMs**. Seven-week-old C57BL/6 mice (Actb-EGFP and mT/mG) and were sprayed with 75% ethanol, and the femurs and tibias were dissected using scissors. The bones were flushed with a syringe filled with RPMI-1640 to isolate bone marrow (BM) cells in a 6-well culture plate. To obtain the BMDCs, the BM cells were cultured in RPMI-1640 supplemented with 10% FBS, 100 U/ml penicillin-streptomycin, 50 μM β-mercaptoethanol, 20 ng/ml murine granulocyte-macrophage colony-stimulating factor (GM-CSF) and 1 ng/ml IL-4 (PeproTech). On day 3, the supernatants were gently removed and replaced with the same volume of medium. On day 6, the non-adherent cells were collected. To obtain BMDMs, BM cells were cultured in DMEM containing 10% FBS, 100 U/ml penicillin-streptomycin, 50 μM β-mercaptoethanol and 20 ng/ml murine macrophage colony-stimulating factor (M-CSF, PeproTech). Four days after seeding the cells, supernatants were replaced with the same volume of medium and the progenitor cells were incubated for an additional 3 days. The attached cells were collected by washing with ice-cold PBS. After obtaining BMDCs and BMDMs, the cells were counted and seeded in confocal dishes 12 h before the next experimental procedure.

**LN-targeting assay**. For LN imaging, animals were sacrificed, and ILNs and ALNs were excised and imaged using a custom-made whole-body optical imaging system 3 h and 6 h injection of FITC-α-melittin-NPs, FITC-α-peptide-NPs and FITC-melittin. The fluorescence imaging of FITC was acquired with filter set (excitation: 469/35 nm; emission: 559/34 nm) and calibrated with an autofluorescence background filter set (excitation: 390/40 nm; emission: 559/34 nm). For the FITC quantification experiments, tumors and other organs were collected, weighed, and mechanically digested in PBS for 5 min. Then, the tissues were sonicated using a sonicator (JY92-IIDN, Scientz, Ningbo) for 30 s at 3 watts of output power.

Following the addition of 10% trichloro-acetic acid in methanol, samples were centrifuged at 12,000 × $g$ for 15 min. The detection of FITC in supernatants was performed using a Bio-Tek Epoch microplate spectrophotometer (Winooski, Vermont, USA).

**Blood smear**. Approximately 100 μl blood samples were collected in Eppendorf tubes containing 5 μl heparin solution (1000 U/ml). Blood smears were made on a microscope slide and fixed for 3 min with Wright-Giemsa staining solution (Solarbio, Beijing). Then, an equal or slightly larger amount of PBS buffer solution (pH = 6.4) was added to the smear and was allowed to stain for an additional 5 min. After staining, the slide was washed with tap water. The morphology of RBCs was observed using a Nikon Ni-E (Nikon, Minato, Tokyo, Japan), and the number of schistocytes was counted manually from four fields of view (FOVs).

**Flow cytometry**. Antibodies to CD45 (Clone: 30-F11, Catalog: 103116/103112/103108), CD3 (Clone: 145-2C11/17A2, Catalog: 100308/100204), CD4 (Clone: RM4-5, Catalog: 100514), CD8 (Clone: 53-6.7, Catalog: 100722), B220 (Clone: RA3-6B2, Catalog: 103223), CD11c (Clone: N418, Catalog: 117316/117324), F4/80 (Clone: BM8, Catalog: 123132), CD11b (Clone: M1/70, Catalog: 101216/101212), Ly-6G (Clone: 1A8, Catalog: 127608/127624), Ly-6C (Clone: HK1.4, Catalog: 128012), NK1.1 (Clone: PK136, Catalog: 108710), CD80 (Clone: 16-10A1, Catalog: 104722), CD86 (Clone: GL-1, Catalog: 105012), IFN-γ (Clone: XMG1.2, Catalog: 505808), TNF-α (Clone: MP6-XT22, Catalog: 506308) and CD16/32 (Clone: 93, Catalog: 101320) were purchased from Biolegend. The fixable viability dye eFluor506 (Catalog: 65-0866-18) were purchased from eBioscience. Cell were isolated from lymph node and tumor as follow: lymph node and tumors were removed using forceps and surgical scissors and weighed. Then they were minced with scissors prior to incubation with 2 mg/ml collagenase IV and 0.2 mg/ml DNase I (Sigma-Aldrich) for 30 min at 37 °C. These tissues were homogenized by repeated pipetting and filtered through a 70 μm cell strainer, and then washed once with complete RPMI to prepare a single-cell suspension. Infiltrating immune cells counts were normalized by tumor mass. For intracellular staining, cell surface antigen staining was firstly performed, and cells were fixed in 0.5 ml/tube Fixation Buffer in the dark for 20 min at room temperature. Then these cells were resuspended in Intracellular Staining Perm Wash Buffer and stained with TNF-α and IFN-γ antibodies. The cell density was analyzed using a micro- capillary flow cytometer (Guava EasyCyte8HT, EMD Millipore Corporation, Billerica, MA, USA). The expression of cell surface markers was analyzed using a CytoFLEX flow cytometer (Beckman Coulter, USA). The data were analyzed using FlowJo software (FlowJo, Ashland, OR, USA).

**Tumor inoculation and therapy**. For the bilateral flank B16F10 tumor model, tumors were implanted by injection of $1 \times 10^5$ cells in the left flank intradermally on day 0 and $7.5 \times 10^4$ cells in the left flank on day 4. Mice were randomized into different treatment groups 7 days after the injection on the left flank and were treated with intratumoral injections of 35 nmol melittin, α-peptide-NPs, and α-melittin-NPs in PBS, with a total volume of 50 μl, as indicated in Fig. 3a. Mice that received an intratumoral injection of PBS were used as the control group. The quantification was based on the peptide content. Tumor size was measured using digital calipers and volumes (mm$^3$) were calculated according to the following formula: $V = 0.5 \times$ length $(L) \times$ width $(W)^2$. In order to analyze the tumor-infiltrating lymphocytes, B16F10 tumor cells were subcutaneously injected on the left $(1 \times 10^5)$ and right flanks $(1.5 \times 10^5)$ of mice on days 0 and 4, respectively. Mice were treated as above. The rare mice that died from tumor burden (usually in PBS and α-peptide-NPs groups) or mice that have no visible tumor in the right flank (usually in melittin and α-melittin-NPs groups) were not used for the analysis.

**Immunofluorescence staining**. For Immunofluorescence analysis, freshly harvested tumor-draining lymph nodes and tumor tissues were quickly rinsed PBS and fixed with 10% neutral buffered formalin (sigma, UK) for 10 h. Tissues were cryoprotected with 30% sucrose in PBS at 4 °C until sinking (usually within 24 h) and then embedded in OCT compound (Sakura, Torrance, CA, USA) before freezing on dry ice. 10 μm-thick tissue sections were obtained with on a Leica CM1950 cryostat (Wetzlar, Germany) and adhered to poly-L-lysine-coated slides. For staining, slides were washed once with PBS before blockage of non-specific binding sites with 2% BSA in PBS for 1 h. Sections were then incubated-within a humidified chamber for 1 h in 2% BSA with primary antibodies specific for F4/80 (Clone: BM8, Catalog: 123132, 1:200), CD11c (Clone: N418, Catalog: 117314, 1:100), CD3 (Clone: 17A2, Catalog: 100240, 1:200), B220 (Clone: RA3-6B2, Catalog: 103226, 1:200), CD4 (Catalog: 562891, 1:200) and CD8 (Clone: 53-6.7, Catalog: 100724, 1:200). After three washes with PBS, sections were imaged using a Zeiss LSM 710 confocal imaging system (Oberkochen, Germany) with a dry 20×/0.8NA objective. The data were analyzed using ImageJ software.

**Cholesterol estimation**. For total cellular cholesterol estimation, cells (B16F10, BMDM, and BMDC) were lysed in PBS containing 2% Triton X-100 for 10 min. After centrifugation (12,000 rpm, 15 min), resulting supernatant was used for detecting cholesterol content with Amplex® Red reagent-based assay (Invitrogen, Carlsbad, CA, USA).

**Cytokine and chemokine quantitation**. Tumor tissues were harvested, and their masses were measured. Then, tissue samples were lysed in lysis buffer (5 μl/mg) containing 50 mM Tris-HCl (pH 7.5), 10% glycerol, 150 mM NaCl, and 1% NP-40 and freshly supplemented with protease inhibitor cocktail (Sigma-Aldrich). Lysates were aliquoted and stored at −80 °C until analysis. Samples were assayed using the LEGENDplex™ mouse inflammation and chemokine panel array (BioLegend) according to the manufacturer's instructions. The data was analyzed with LEGENDplex software (BioLegend).

**Statistical analysis**. A one-way analysis of variance (ANOVA) followed by a post hoc test was used for multiple group comparisons. Data for survival were analyzed by a log-rank (Mantel-Cox) test. All statistical analyses were performed with GraphPad Prism 7.2 (GraphPad Software, CA, USA). Data were presented as mean ± SEM. Differences were considered statistically significant at *$P < 0.05$, **$P < 0.01$, ***$P < 0.001$ and ****$P < 0.0001$. The numbers of animals included in the study are discussed in each figure.

**Reporting summary**. Further information on research design is available in the Nature Research Reporting Summary linked to this article.

## Data availability

The authors declare that the main data supporting the findings of this study are available within the Article and the Supplementary Information or available from the corresponding author upon reasonable request. The source data underlying Figs. 1c, d, g, 2d, 3c-f, 4b, c, 5b-e and 6b, c, e and Supplementary Figs. 2b, 3a, 4, 5a–c, 6–8, 11–12, and 13a–c are provided with the paper as a Source Data file.

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

## Acknowledgements

We thank the Optical Bioimaging Core Facility of WNLO-HUST and the Research Core Facilities for Life Science (HUST) for the support in data acquisition, and the Analytical and Testing Center of HUST for the facility support. This work was supported by the National Science Fund for Distinguished Young Scholars (81625012), the Science Fund for Creative Research Groups of the National Natural Science Foundation of China (61721092), the Fundamental Research Funds for the Central Universities (2019kfyXMBZ022), Program for HUST Academic Frontier Youth Team (Zhang, Z.H.), and the Innovation Fund of WNLO.

## Author contributions

X.Y. and Y.D. under direction of Z.Z. designed and performed experiments, analyzed the data, and wrote first version of manuscript. Y.Z., S.Q., L.L., and L.L performed part of experiments. Q.L. and Z.Z. designed experiments and supervised the study. X.Y. and Z.Z. wrote the manuscript. All authors contributed to the critical revision of the manuscript and approved the final version.

## Competing interests

The authors declare no competing interests.
