## [Peer Review File · Nature Communications]

Reviewers' comments:

Reviewer #2 (Remarks to the Author): Expert in nanoparticles and immunology

In this paper, "Melittin-lipid nanoparticles as a whole-cell in situ nanovaccine that targets to lymph node and elicits a systemic antitumor immune response," the authors describe an approach for a nanovaccine that can induce the release of tumor antigens and stimulate a systemic immune response that delays or reverses tumor growth in a mouse melanoma model. Their strategy combines the lymph node-homing properties of small nanoparticles (NPs) with the immunomodulatory effects of melittin, a peptide found in bee venom. The authors have previously published on the properties and anti-tumor activity of melittin-based nanoparticles, and this study extends the work to explore the biodistribution and immune-related effects of the NPs.

The authors argue that current nanovaccine approaches rely on complex or clinically infeasible approaches for identifying, extracting, and loading tumor-associated antigens (TAAs) into nanoparticles. To overcome these limitations, this strategy aims to develop a nanoparticle formulation of an immunostimulatory peptide- melittin- that accumulates in the lymph nodes, priming antigen-presenting cells (APCs) to attack tumor cells upon recognition of TAAs.

Specifically, the authors chose intratumoral delivery of a melittin nanoparticle to 1) promote antigen release from the primary tumor, 2) accumulate in lymph nodes and prime APCs to recognize the TAAs, and 3) induce cytotoxic immune action against tumor cells.

The authors address the question of NP biodistribution and uptake by APCs using fluorescence microscopy and flow cytometry of FITC-labeled nanoparticles in lymph nodes extracted from mice. The cytotoxicity of the α -melittin-NPs was then compared to that of free melittin in vitro in normal bone marrow-derived APCs and melanoma cells. The authors used a bilateral flank model to perform efficacy studies, supporting their initial hypothesis that the α -melittin-NPs can inhibit tumor growth more effectively than free melittin. Modulation of cytokine levels in the LNs and contralateral tumor was assessed by flow cytometry, suggesting that enhanced vaccine efficacy was a product of cellular immunity.

General assessment:

This work is of high importance and should be published, but requires significant clarifications and discussion of the issues in the major comments before publication.

Major comments:

-Figure 1f: the values of the per cents in the text (lines 150-151) do not match the values of the per cents in the gates in the figure. The authors should clarify the definition of the gates in each context.

-What is the rationale for inoculating the two tumors 4 days apart instead of simultaneously? What is the rationale for the in vivo doses chosen? In some instances, 20 nmol of melittin are used, and in others, 35 nmol are used. Authors should provide the rationale for inoculation method, dose and frequency, as well as the volume of the tumors upon injection and their reasoning for the volume of 50 μ L used for intratumoral injection.

-In supplementary figure 7, the fluorescence analysis and quantification indicate that the organs retained an undetectable level of FITC, but how is this directly correlated to the melittin content? LC-MS experiments can help verify that the melittin is also absent in these organs. The authors should provide the ratio of FITC to melittin or peptide or nanoparticle to help clarify this correlation as well.

-A discussion of the incomplete response observed in vivo is necessary. A discussion of why some of the α -melittin-NP-treated animals experience complete tumor regression and others' tumors continued to grow at an exponential rate should be provided. The authors should further discuss possible resistance issues or potential reasons for failed or incomplete responses.

-In the introduction, it will be helpful to include background information on the levels of possible tumor-associated antigens in melanoma. In the discussion section, the translatability of this work can be expounded on by describing other cancer types in which this approach may be applicable. The authors should include a discussion of some ways that this strategy can be extended if an intratumoral injection is infeasible.

-In the introduction, the authors mention that current practices for loading whole-cell tumor

antigens are inconvenient. It can be useful to mention the ease/cost of melittin synthesis or purification in the discussion section as a comparison.

-Since melittin is a major component of bee venom, and some individuals are allergic to bee venom, will this be a consideration when stratifying patients if this strategy is to be translated to the clinic? The authors should address the issue of melittin as a general allergen.

-Fig. S5a is difficult to interpret, as the fluorescence intensity of the "fluorescent model antigens" look similar in the PBS control image compared with that of the α -melittin-NP. A clarification is needed.

-The statement in line 192-193 regarding the idea that the melittin "induces release of whole tumor cell antigens" has not yet been fully demonstrated at that point in the paper. This can be moved to later in the manuscript, after the appropriate results have been shown.

Minor comments

-The figure numbers are not sequential in a few cases. For instance, Fig. 1d and 1e are discussed prior to 1b and 1c.

-The cancer type used for in vivo studies can be mentioned in the abstract and earlier upon introduction of the cell line name, clarifying that it is melanoma.

-The middle panels of Fig. 1a were not mentioned in the main text.

-In Fig. 1b, there is no indication that these agents are FITC-conjugated.

-The MTS assay results in Fig. 2 c, d can be described clearly by the IC50 values.

-A few of the word choices can be improved (i.e. "fantastic" in line 55, "just like killing two birds with one stone" in line 116).

Reviewer #3 (Remarks to the Author): Expert in melittin and nanoparticles

These authors are investigating the ability of peptide-lipid nanoparticles to enhance the presentation of tumor antigens. This particular type of nanoparticle has been available for some time and is known to be of small size (10–100 nm) which is desirable to provide good tissue access. This manuscript presents the preparation only briefly and does not address the particle composition and structure over the size range that was used. In the first experiments presented fluorescein labeled melittin and NPs were used to demonstrate that formulation into nanoparticles was critical to permit transit to lymph nodes. However what is not presented is the comparative activity of the FITC substituted nanoparticles. This could have been accomplished using the effect on contra lateral tumors but that was not tested with the FITC nanoparticles.

The authors show that cultured tumor cells are more sensitive to α -melittin NPs than isolated cultured antibody presenting cells. It is not clear to this reviewer that this experiment contributes to the authors hypothesis. There is no development of the basis for this difference. This cultured cell protocol could be used to determine if the FITC labeled NPs are as active as the underivatized probes that were used. The significance of the tFRFP expressing cells is not clear in that if apoptosis is the mechanism of cultured cell demise the expressed antigen would be lost in the small vesicles formed during apoptosis. Furthermore the authors use TUNEL assays to indicate that apoptosis is prominent in cultured cells exposed to the melittin protocols. The TUNEL assay can be applied to tissue, cultured cells and in flow cytometry. Data for the initial tumor, draining lymph nodes, distant tumor and induced lymphocytes would speak directly to the mechanism of the "one stone – two birds" hypothesis.

The major data in this manuscript is a series of bilateral flank tumor experiments that are intended to verify "the whole cell vaccine" importance to melittins action on cancer cells. The results in this manuscript are striking

and strongly support a role for melittin nanoparticles in cancer therapy. The contra lateral tumor specific and prolonged suppression of tumor growth that the authors present are strong indications that this protocol has merit. The authors then address the role of cellular immunity verses serum antibody response. The cellular response was robust and specific. The IGG response seemed much less dramatic in that the melittin activated response was only twice that of the controls. Then the authors investigate the tumor environment resulting from melittin promoted lymphocyte infiltration. These experiments were presented as a single time point making it difficult to assess the chronological importance of the melittin induced changes. This work is carefully planned and strongly suggestive but incomplete in at least three ways. This limits the potential for mechanistic interpretations that can be supported. This is reflected in their discussion which re-presents the phenomena reported in the Results.

Note

- In the Discussion the authors refer to work by Dezfuli but I did not find a reference to this work in the manuscript.

Paul H. Schlesinger, MD, PhD

Point-by-point responses to the reviewers' comments

We would like to express our sincere thanks to the editor and all reviewers for their critical and constructive comments. We have performed substantial additional experiments to respond point-by-point to their concerns. We feel that their comments have helped us to significantly improve and strengthen the manuscript, as well as clarify some of the important issues of our work. We hope that the revision has addressed their major concerns. The revised sections are marked in red, and the data that were not added to the manuscript and are presented only below in the response to the reviewers' concerns are labelled with **Figs. R1-4**.

Reviewer #2 (Remarks to the Author): Expert in nanoparticles and immunology

Comments to authors:

In this paper, "Melittin-lipid nanoparticles as a whole-cell in situ nanovaccine that targets to lymph node and elicits a systemic antitumor immune response," the authors describe an approach for a nanovaccine that can induce the release of tumor antigens and stimulate a systemic immune response that delays or reverses tumor growth in a mouse melanoma model. Their strategy combines the lymph node-homing properties of small nanoparticles (NPs) with the immunomodulatory effects of melittin, a peptide found in bee venom. The authors have previously published on the properties and anti-tumor activity of melittin-based nanoparticles, and this study extends the work to explore the biodistribution and immune-related effects of the NPs.

The authors argue that current nanovaccine approaches rely on complex or clinically infeasible approaches for identifying, extracting, and loading tumor-associated antigens (TAAs) into nanoparticles. To overcome these limitations, this strategy aims to develop a nanoparticle formulation of an immunostimulatory peptide- melittin- that accumulates in the lymph nodes, priming antigen-presenting cells (APCs) to attack tumor cells upon recognition of TAAs. Specifically, the authors chose intratumoral delivery of a melittin nanoparticle to 1) promote antigen release from the primary tumor, 2) accumulate in lymph nodes and prime APCs to recognize the TAAs, and 3)

induce cytotoxic immune action against tumor cells.

The authors address the question of NP biodistribution and uptake by APCs using fluorescence microscopy and flow cytometry of FITC-labeled nanoparticles in lymph nodes extracted from mice. The cytotoxicity of the α -melittin-NPs was then compared to that of free melittin in vitro in normal bone marrow-derived APCs and melanoma cells. The authors used a bilateral flank model to perform efficacy studies, supporting their initial hypothesis that the α -melittin-NPs can inhibit tumor growth more effectively than free melittin. Modulation of cytokine levels in the LNs and contralateral tumor was assessed by flow cytometry, suggesting that enhanced vaccine efficacy was a product of cellular immunity.

This work is of high importance and should be published, but requires significant clarifications and discussion of the issues in the major comments before publication.

Major comments:

1. Figure 1f: the values of the percents in the text (lines 150-151) do not match the values of the percents in the gates in the figure. The authors should clarify the definition of the gates in each context.

Response:

We thank the reviewer for their constructive comments. In the original manuscript, the percent values in the text represent the average of several samples, and the values in the gates in Fig. 1f indicate the percentage of the representative sample. According to the reviewer's suggestion, the values in the gates in Fig. 1f are shown as the mean \pm SEM in our revised manuscript.

2. What is the rationale for inoculating the two tumors 4 days apart instead of simultaneously? What is the rationale for the in vivo doses chosen? In some instances, 20 nmol of melittin are used, and in others, 35 nmol are used. Authors should provide the rationale for inoculation method, dose and frequency, as well as the volume of the tumors upon injection and their reasoning for the volume of 50 μ L used for intratumoral injection.

Response:

We thank the reviewer for their detailed comments. The bilateral flank tumor model is often used to characterize local and abscopal effects of intratumoral therapy, and the time interval for inoculating two tumors is not standard. According to the corresponding references, tumor cells could be injected in the right flank (contralateral tumor) on days 0 (simultaneously)¹, 2^{2,3}, 3⁴, 4⁵⁻⁷, and 6⁸. In our study, the contralateral tumor was implanted in 4-day intervals according to the majority of existing literature.

Because of the different ratio of fluorescein to melittin or the peptide-based nanoparticle, we chose different doses in different experiments. Specifically, we used 20 nmol (quantification was based on the FITC content) of FITC-labelled melittin, α -peptide-NPs, and α -melittin-NPs to evaluate the distribution in **Fig. 1a-e and Supplementary Figs. 1-3, and 11**. Because of functional role of melittin, we used 35 nmol (quantification was based on the functional peptide content) of melittin, α -peptide-NPs, and α -melittin-NPs to evaluate the activation and treatment effect in **Fig. 1f,g, Figs. 3-6, and Supplementary Figs. 4, 9, 10, 12 and 13**. According to the reviewer's suggestion, we have added a related description of the dose in the corresponding position, and the data for the volume of the tumors upon injection were provided in the Source Data file.

We modified the manuscript as follows:

The manuscript and supplementary materials (Figure legends: Fig. 4a, Supplementary Figs. 10 and 11) now states the following: “35 nmol of α -melittin-NPs (quantification was based on the peptide content) in PBS”, “35 nmol of melittin, α -peptide-NPs, and α -melittin-NPs (quantification was based on the peptide content)”, and “20 nmol of FITC- α -melittin-NPs (quantification was based on the FITC content)”.

3. In supplementary figure 7, the fluorescence analysis and quantification indicate that the organs retained an undetectable level of FITC, but how is this directly correlated to the melittin content? LC-MS experiments can help verify that the

melittin is also absent in these organs. The authors should provide the ratio of FITC to melittin or peptide or nanoparticle to help clarify this correlation as well.

Response:

We thank the reviewer for their constructive suggestion. Fluorescein isothiocyanate (FITC) is an amine reactive derivative of fluorescein dye that has been extensively used to label siRNA, peptides and proteins for analyzing distribution⁹⁻¹². In our original manuscript (**Supplementary Fig. 7**), FITC were extracted from organs by ultrasonic extraction with methanol and analyzed by a microplate reader. Essentially, the FITC content was quantified with fluorescence (excitation: 495 nm; emission: 525 nm). Our data showed that organs seemed to retain an undetectable level of FITC, especially in the liver, kidney and heart. Organs emitted autofluorescence signals because of the presence of endogenous fluorochromes, such as nicotinamide adenine dinucleotide (NADH), riboflavin, aromatic amino acids, etc.¹³. Riboflavin is the most likely fluorophore inducing an emission signal in the range of 520-540 nm and is rich in the liver, kidney and heart^{14,15}. We speculate that the organ autofluorescence interferes with the detection of FITC. However, by subtracting the organ autofluorescence signal, we still detected tiny amounts of FITC in the organs (**Supplementary Fig. 11**). Therefore, we concluded that a very small proportion of FITC- α -melittin-NPs entered the bloodstream. As discussed in the legend in Supplementary Fig. 11, α -melittin-NPs are mainly restricted to the injected tumor and tumor-draining LNs.

Supplementary Figure 11. α -melittin-NPs are mainly restricted to the injected tumor and tumor-draining LNs. Mice ($n = 3$) were inoculated with B16F10 cells in the left and right flanks on days 0 and 4, respectively. 20 nmol of FITC- α -melittin-NPs (quantification was based on the FITC content) were intratumorally administered when the tumor size reached 100 mm³. Tumors and other organs were collected, weighed, and mechanically digested in PBS for 5 mins. Then, the tissues were sonicated using a sonicator for 30 seconds at 3 watts of output power. Following the addition of 10% trichloro-acetic acid in methanol, samples were centrifuged at 12,000 g for 15 min. The detection of FITC in supernatants was performed via fluorescence (excitation: 495 nm; emission: 525 nm) using a microplate reader.

To confirm the correlation between FITC and melittin, we added the fast protein liquid chromatography (FPLC) data in Fig. R1. As you can see, the absorption curves at 280 nm (represents peptides) and 495 nm (represents FITC) of FITC-melittin, FITC- α -peptide-NPs, and FITC- α -melittin-NPs basically trended the same, indicating that FITC was successfully conjugated to the primary amines of peptide.

Figure. R1. The FPLC profile of FITC-labelled melittin or nanoparticles. (a) FITC-melittin; (b) FITC- α -peptide-NP; (c) FITC- α -melittin-NP.

In addition, the ratio of FITC to peptide was determined by spectrophotometric analysis and calculated using the following formula according to the manufacturer's instructions:

$$\text{Molar F/P} = \frac{\text{MW}}{389} \times \frac{A_{495}/195}{[A_{280} - (0.35 \times A_{495})]/E^{0.1\%}}$$

where MW is the molecular weight of the peptide, 389 is the molecular weight of FITC, 195 is the absorption $E^{0.1\%}$ of bound FITC at 490 nm at pH 13.0, $(0.35 \times A_{495})$ is the correction factor due to the absorbance of FITC at 280 nm, and $E^{0.1\%}$ is the absorption at 280 nm of a protein at 1.0 mg/ml. The ratio of FITC to peptide in FITC-melittin, FITC- α -peptide-NPs and FITC- α -melittin-NPs was 0.17, 0.20 and 0.40, respectively (**Table. R1**).

Table. R1. The ratio of FITC to peptide

	A495	A280	E ^{0.1%}	F/P
FITC-melittin	3.25	1.33	0.70	0.17
FITC- α -peptide-NP	3.57	1.80	0.92	0.20
FITC- α -melittin-NP	3.91	1.84	0.70	0.40

4. A discussion of the incomplete response observed in vivo is necessary. A discussion of why some of the α -melittin-NP-treated animals experience complete tumor regression and others' tumors continued to grow at an exponential rate should be provided. The authors should further discuss possible resistance issues or potential reasons for failed or incomplete responses.

Response:

We thank the reviewer for their constructive suggestion. In the revised manuscript, we have added a discussion of the incomplete response.

We modified the manuscript as follows:

Page 15, line 411 of the manuscript (Discussion) now states the following: “However, the incomplete response occurred in both α -melittin-NP and melittin groups. Tumor progression involves the co-evolution of neoplastic cells together with tumor microenvironment, and heterologous cell types within tumors can actively influence the therapeutic response and shape resistance, even in cases in which immune cell actively drive the initial response to targeted therapies¹⁶. In some individuals, CD8⁺ T cells infiltration may be concomitant with the elevated level of T cell inhibitory receptors, such as T lymphocyte-associated antigen-4 (CTLA-4) and programmed death 1 (PD1), leading to the emergence of T cell exhaustion. For example, while oncolytic virotherapy induced the infiltration of activated lymphocytes in tumors, the antitumor effect was unable to lead to complete tumor regression because of treatment-induced adaptive immune resistance manifested by upregulation of CTLA-4 or PD1^{5,17,18}. Therefore, we hypothesize that the incomplete response

induced by α -melittin-NPs in certain individuals can be improved by combination strategies using checkpoint inhibitors”.

5. In the introduction, it will be helpful to include background information on the levels of possible tumor-associated antigens in melanoma. In the discussion section, the translatability of this work can be expounded on by describing other cancer types in which this approach may be applicable. The authors should include a discussion of some ways that this strategy can be extended if an intratumoral injection is infeasible.

Response:

We thank the reviewer for their very constructive suggestion. In the revised manuscript, we have added background information on tumor-associated antigens in melanoma and a discussion about the translatability potential of this strategy.

We modified the manuscript as follows:

Page 3, line 60 of the manuscript (Introduction) now states the following: “However, the number of available TAAs, which are a key component of vaccines, to load onto the nanovaccines are very limited for most types of cancers. **Although several TAAs for melanoma have been defined, such as melanocyte differentiation antigens (MDAs),** the immunogenicity of TAAs is highly variable among individuals, and TAAs can undergo immune-editing to escape immune recognition during tumor development^{19,20}. **In addition, neoantigens that arise as a consequence of tumor-specific mutations have been proved to be of particular relevance to tumor control^{21,22}, but the prediction of individualized neoantigens is mainly restrained by sophisticated technology^{23,24}.**”

Page 14, line 401 of the manuscript (Discussion) now states the following: “**Because intratumoral injection allows much higher concentrations of immunostimulatory products in the tumor microenvironment than systemic administration, intratumoral treatment is still popular²⁵⁻²⁸. However, any intratumoral therapy requires access to the tumor site. The accessibility of primary melanomas and subcutaneous breast tumors provide interesting examples. For deep-seated malignant tumors, this strategy must turn to the guidance with B-ultrasonography or CT.**”

6. *In the introduction, the authors mention that current practices for loading whole-cell tumor antigens are inconvenient. It can be useful to mention the ease/cost of melittin synthesis or purification in the discussion section as a comparison.*

Response:

We thank the reviewer for their constructive suggestion. In the revised manuscript, we have added a discussion about the ease/cost of melittin-lipid nanoparticle synthesis and purification.

We modified the manuscript as follows:

Page 15, line 424 of the manuscript (Discussion) now states the following: “**Current practices for LN-targeted whole-cell nanovaccines involve the preparation of tumor tissues by surgery as well as antigen loading in vitro. These steps require a substantial time commitment and may cause potential contamination. In our study, intratumoral direct injection of α -melittin-NPs turned the tumor into a vaccine factory without any manual manipulations in vitro. In addition, the α -melittin-NPs components include phospholipid, cholesterol oleate and peptide. Both phospholipid and cholesterol oleate have perfect biocompatibility and are relatively inexpensive. The last few years have also seen a remarkable decrease in the cost of peptide synthesis due to the cost of raw materials and technical improvements in peptide reverse phase flash chromatography²⁹. More importantly, the application of nanotechnology and intratumoral injection greatly decreased the required dosage of melittin peptide, generating further cost reductions.**”

7. *Since melittin is a major component of bee venom, and some individuals are allergic to bee venom, will this be a consideration when stratifying patients if this strategy is to be translated to the clinic? The authors should address the issue of melittin as a general allergen.*

Response:

We thank the reviewer for their detailed suggestion. Although melittin is a major component of bee venom, it is a relatively weak allergen³⁰. It was reported that the

best characterized allergen of bee venom is phospholipase A2³¹⁻³³. In addition, the prevalence of systemic allergic reactions to Hymenoptera stings ranges from 0.3 to 7.5% in adults and up to 3.4% in children³⁴. If allergic reactions occur, the effective treatment to prevent further systemic sting reaction is venom immunotherapy (VIT). The European Academy of Allergy and Clinical Immunology's (EAACI) Taskforce prepared a detailed guideline on VIT. According to the formal systematic review and meta-analysis of a large sample, VIT proved to be highly effective and safe, and no fatalities were recorded³⁴.

8. *Fig. S5a is difficult to interpret, as the fluorescence intensity of the “fluorescent model antigens” look similar in the PBS control image compared with that of the α -melittin-NP. A clarification is needed.*

Response:

We thank the reviewer for their constructive comments. In the original Supplementary Fig. 5a, we directly overlapped the fluorescence channels of the fluorescent model antigen and TUNEL, making the phenomenon of disappearance of fluorescent model antigens not too obvious. In the revised manuscript, we have re-processed the images in Fig. S5a by separating the fluorescence channels. Meanwhile, we also subtracted the background fluorescence in the TUNEL channel. As you can see in Supplementary Fig. 9 below, because of the function of disrupting cell membranes, both melittin and α -melittin-NPs induced the disappearance of fluorescent model antigens in the tumor section (area marked by a white dotted outline), indicating that the tumor antigens had been released.

Supplementary Figure. 9. Representative immunofluorescence imaging of necrotic/apoptotic tumor cells induced by melittin and α -melittin-NPs. The white dotted outline indicates the area of necrosis/apoptosis. Scale bar, 10 μ m.

9. The statement in line 192-193 regarding the idea that the melittin “induces release of whole tumor cell antigens” has not yet been fully demonstrated at that point in the paper. This can be moved to later in the manuscript, after the appropriate results have been shown.

Response:

We thank the reviewer for their constructive suggestions. As shown above (Supplementary Fig. 9), melittin itself has the ability to kill B16F10 tumor cells and induce the release of whole tumor cell antigens. When melittin was loaded onto the scaffold of α -peptide-NP, it could form an ultrasmall melittin-lipid nanoparticle, named as α -melittin-NP. This α -melittin-NP maintained the ability of melittin to directly induce tumor cell necrosis/apoptosis. More importantly, the α -melittin-NP has the required size for an optimal LN-targeted nanovaccine that can efficiently drain into lymphatic capillaries and lymph nodes.

Minor comments:

1. The figure numbers are not sequential in a few cases. For instance, Fig. 1d and 1e are discussed prior to 1b and 1c.

Response:

We thank the reviewer for their constructive suggestions. According to the reviewer's suggestions, we have described **Fig. 1b** and **1c** earlier in the revised manuscript.

2. The cancer type used for in vivo studies can be mentioned in the abstract and earlier upon introduction of the cell line name, clarifying that it is melanoma.

Response:

We thank the reviewer for their constructive suggestions. According to the reviewer's suggestions, we mentioned the cancer type earlier in the revised manuscript.

We modified the manuscript as follows:

Page 2, line 30 of the manuscript (Abstract) now states the following: "Time-lapse imaging showed that α -melittin-NPs displayed low toxicity to APCs but maintained the toxicity of melittin to **B16F10 melanoma cells**".

3. The middle panels of Fig. 1a were not mentioned in the main text.

Response:

We thank the reviewer for their critical comments. In our revised manuscript, we have mentioned the middle panels of **Fig. 1a**.

We modified the manuscript as follows:

Page 5, line 130 of the manuscript (Results: α -melittin-NPs enhance the LN uptake of melittin and activate APCs) now states the following: “Wide-field fluorescence imaging data showed that α -melittin-NPs **as well as the α -peptide-NPs scaffold** led to their substantial accumulation in inguinal LNs (ILNs) and axillary LNs (ALNs) (**Fig. 1a, lower and middle panels**)”.

4. The MTS assay results in Fig. 2 c, d can be described clearly by the IC50 values.

Response:

We thank the reviewer for their constructive suggestions. According to the reviewer’s suggestions, we have described IC50 values in the revised manuscript.

We modified the manuscript as follows:

Page 7, line 173 of the manuscript (Results: α -melittin-NPs have more cytotoxic effects on tumor cells than on APCs) now states the following: “**Compared with the half-maximal inhibitory concentrations (IC50s) of free melittin (2.61 μ M for BMDCs and 0.97 μ M for BMDMs), those of α -melittin-NPs appeared to have decreased cytotoxicity to BMDCs and BMDMs, as indicated by significantly increased IC50 values (30.41 μ M for BMDCs and 22.82 μ M for BMDMs)(Supplementary Fig. 5)**”.

5. A few of the word choices can be improved (i.e. “fantastic” in line 55, “just like killing two birds with one stone” in line 116).

Response:

We thank the reviewer for their constructive suggestions. According to the reviewer’s suggestions, the word “fantastic” has been changed to “**great**” in the revised manuscript. In addition, “killing two birds with one stone” is an old proverb meaning that we can achieve two things in a single action, and has been used in research papers³⁵⁻³⁸.

Reviewer #3 (Remarks to the Author): Expert in melittin and nanoparticles

Comments to authors:

This work is carefully planned and strongly suggestive but incomplete in at least three ways. This limits the potential for mechanistic interpretations that can be supported. This is reflected in their discussion which re-presents the phenomena reported in the Results.

Major comments:

1. These authors are investigating the ability of peptide–lipid nanoparticles to enhance the presentation of tumor antigens. This particular type of nanoparticle has been available for some time and is known to be of small size (10–100 nm) which is desirable to provide good tissue access. This manuscript presents the preparation only briefly and does not address the particle composition and structure over the size range that was used. In the first experiments presented fluorescein labeled melittin and NPs were used to demonstrate that formulation into nanoparticles was critical to permit transit to lymph nodes. However, what is not presented is the comparative activity of the FITC substituted nanoparticles. This could have been accomplished using the effect on contra lateral tumors but that was not tested with the FITC nanoparticles.

Response:

We thank the reviewer for their detailed comments. According to the reviewer's suggestions, we provided additional detailed information about the preparation of nanoparticles in the method sections. In addition, we compared the basic characteristics (such as surface charge, particle size, and cytotoxic activity) of fluorescently labelled or unlabelled α -melittin-NPs in **Fig. R2**. Apolipoprotein A1 (ApoA1) is a main protein component in high-density lipoprotein (HDL) particles. In spherical HDL particles, the amphipathic α -helix in ApoA1 can interact with phospholipids and cover ~80% of the total surface, underlining the major role of this apolipoprotein in stabilizing HDL particle structure and shape^{39,40}. Therefore, the ApoA1-packing density increase indicates a reduction in the spherical particle size⁴¹.

The melittin nanoparticles (α -melittin-NPs) consisted of phospholipid [1,2-dimyristoyl-sn-glycero-3-phosphocholine (DMPC)], cholesterol oleate (CO), and α -melittin, among which α -melittin was designed by hybridizing melittin and an ApoA1-mimetic peptide (D4F, noted as α -peptide). More importantly, α -melittin displayed a strong α -helical configuration. As shown in our previous study, the interaction between the α -melittin network and the lipid monolayer resulted in nanoparticles that appeared spherical in shape and possessed a small particle size (10~20 nm)⁴². Therefore, α -melittin is a crucial factor in precisely controlling the structure and size of the α -melittin-NPs. In the methods section, we introduced the preparation of the nanoparticles as well as the function of the α -melittin peptide in detail.

Fluorescein isothiocyanate (FITC) is an amine reactive derivative of fluorescein dye that has been extensively used to label siRNA, peptides and proteins⁹⁻¹². Due the fluorescence quenching of FITC during time-lapse imaging and fluorescence interference of enhanced green fluorescent protein (EGFP, BMDMs and BMDCs were isolated from Actb-EGFP C57BL/6 mice), FITC labelled NPs cannot be used in the cultured cell protocol (Fig.2). In our study, FITC was conjugated to the primary amines of peptides to analyze the distribution characteristics of α -melittin-NPs rather than the vaccine effect. Given the possible influences of labelling, we compared the basic characteristics (such as surface charge, particle size, and cytotoxic activity) of FITC-labelled or unlabelled α -melittin-NPs. The FPLC profile, in which the 280 nm and 495 nm absorption curves represented the peptide and FITC contents, respectively, showed that the retention times of the main peaks of α -melittin-NPs (62.58 min) and FITC- α -melittin-NPs (62.81 min) were not significantly different (**Fig. R2a**). Dynamic light scattering (DLS) data indicated that FITC- α -melittin-NPs (13.6 ± 0.72 nm) had a slightly smaller size than unlabelled α -melittin-NPs (15.41 ± 0.6 nm) but both were smaller than 100 nm (**Fig. R2b**). This result means that both have the required sizes to efficiently drain into lymphatic capillaries and lymph nodes. In addition, the zeta potential also decreased after FITC labelling (**Fig. R2c**). However, despite that, both formulations had similar killing effects on B16F10 cells (**Fig. R2d**).

In our previous study, we found that melittin was buried in the phospholipid monolayer, which avoids the impact of FITC on melittin. It is worth noting that seven C-terminal residues (DWFKAFY) of α -peptide (ApoA1-mimetic peptide in NPs) are not involved in lipid-binding status and are exposed outside the particle⁴³, and the lysine (primary amine) in the seven residues is most likely the binding site of FITC. Not surprisingly, therefore, the conjugation of FITC did not affect the cytotoxic activity of melittin. Above all, FITC labelling has no remarkable influence on the size and cytotoxicity of α -melittin-NPs except the zeta potential of intact NPs.

We modified the manuscript as follows:

Page 17, line 475 of manuscript (Methods: Synthesis of nanoparticle) now states the following: “ Subsequently, the mixture was sonicated for 1 h at 48 °C. α -melittin (0.19 μ mol, hybridized peptide) or α -peptide (0.87 μ mol, an ApoA1-mimetic peptide) was dissolved in PBS and added to the lipid emulsion, and then stored overnight at 4 °C. During the incubation time, the interaction between amphipathic α -helix in peptide (α -melittin and α -peptide) and the lipid monolayer resulted in nanoparticles that appeared spherical in shape and possessed a small particle size. Meanwhile, the concentration of peptide increase means the reduction in the spherical size, and a ratio of 0.5:1 is the optimal weight ratio of peptide to lipid.”

Figure. R2. Comparing characteristics of α -melittin NPs and FITC- α -melittin NPs. (a) FPLC profiles of α -melittin NPs (left panel) and FITC- α -melittin NPs (right panel). **(b)** Size distributions of α -melittin NPs (left panel) and FITC- α -melittin NPs (right panel). **(c)** Zeta potential measurements of α -melittin NPs and FITC- α -melittin NPs. **(d)** Cytotoxicity of α -melittin NPs and FITC- α -melittin NPs to B16F10-mAmetrine (blue). Red indicates PI. Concentration: 10 μ M. All scale bars represent 20 μ m. Data are shown as the mean \pm SEM. ****** P < 0.01, as analyzed by Student's t test (two tailed).

2. The authors show that cultured tumor cells are more sensitive to α -melittin NPs than isolated cultured antibody presenting cells. It is not clear to this reviewer that this experiment contributes to the authors hypothesis. There is no development of the basis for this difference. This cultured cell protocol could be used to determine if the FITC labeled NPs are as active as the underivatized probes that were used. The significance of the tRFEP expressing cells is not clear in that if apoptosis is the mechanism of cultured cell demise the expressed antigen would be lost in the small vesicles formed during apoptosis. Furthermore, the authors use TUNEL assays to indicate that apoptosis is prominent in cultured cells exposed to the melittin protocols. The TUNEL assay can be applied to tissue, cultured cells and in flow cytometry. Data for the initial tumor, draining lymph nodes, distant tumor and induced lymphocytes would speak directly to the mechanism of the “one stone – two birds” hypothesis.

Response:

We thank the reviewer for their detailed comments. According to the reviewer’s suggestions, we added new data in Fig. 2d, e and Supplementary Fig. 6-8.

We modified the manuscript as follows:

Page 7, line 177 of manuscript (Results: α -melittin-NPs have more cytotoxic effects on tumor cells than on APCs) now states the following: “**Because negatively charged phosphatidylserine and O-glycosylated mucins are overexpressed in the plasma membrane of many cancer cells, thereby causing these membranes to carry a slightly higher net negative charge than those of normal eukaryotic cells⁴⁴**, we speculated that the reason for the differential killing effects of α -melittin-NPs is probably related to membrane potential-mediated cellular binding. To further confirm this hypothesis, the cell membrane potential was measured using a Malvern Zetasizer Nano ZS90 instrument. The results showed that the zeta potential of the B16F10 cell membrane (-27.65 ± 0.93 mV) was more negative than that of BMDCs (-14.64 ± 1.87 mV) and BMDMs (-10.78 ± 1.57 mV) (Supplementary Fig. 6). Next, we compared the cellular-binding ability of α -melittin-NPs to these three different cells using flow cytometry. The results revealed that the B16F10 cells captured a dramatically greater

amount of FITC- α -melittin-NP than BMDMs at various concentrations during 1 h and 3 h incubation (**Fig. 2d and Supplementary Fig. 7**). We also noted that there was no difference in the mean fluorescent intensity (MFI) of FITC between B16F10 cells and BMDCs during 1 h incubation (except 1.25 μ M) (**Supplementary Fig. 7**), but as the incubation time was prolonged to 3 h and the concentration increased, B16F10 cells also displayed a significantly higher MFI value than BMDCs (**Fig. 2d**).

To observe the cellular distribution of FITC- α -melittin-NPs in these three different cells in vitro, BMDCs and BMDMs were isolated from mT/mG mice that express a strong red fluorescence protein (tdTomato) in the membrane systems (plasma membrane, lysosome, etc) of all cell types. Confocal imaging data showed that FITC- α -melittin-NPs displayed remarkably stronger fluorescent intensity in B16F10 cells than in BMDCs and BMDMs after incubation for 3 h in a 10 μ M concentration (**Fig. 2e**).

More interestingly, the FITC- α -melittin-NPs were mainly distributed in the cell membrane of B16F10 cells, but they were distributed in the intracellular membranes of BMDCs and BMDMs. This finding means that FITC- α -melittin-NPs quickly move through endocytosis to the intracellular membranes after binding, making APCs resistant to the plasma membrane permeabilization-dependent necrosis.”

Page 13, line 368 of manuscript (Discussion) now states the following: “Although the lytic activity of melittin is mainly associated with its ability to disturb cell membrane integrity by incorporating into phospholipid bilayers^{45,46}, nanoparticle-delivered melittin might elicit apoptosis after trafficking to intracellular membranes via activation of the intrinsic pathway¹¹. Therefore, the different spatial distribution of α -melittin-NPs in B16F10 cells and APCs (BMDCs and BMDMs) seem to not explain simply the differential killing effects. A plausible possibility is the existence of a mechanism of anti-apoptosis. After all, necrosis is mainly an irreversible event, but apoptosis can be regulated. We also previously found that Birc5 (also known as survivin), a member of the inhibitor of apoptosis (IAP) gene family encoding negative regulatory proteins that prevent apoptotic cell death, was upregulated in liver sinusoidal endothelial cells (LSECs, another APC type in the liver) after the

administration of α -melittin-NPs⁴⁷. However, it is worth noting that α -melittin-NPs also exhibited cytotoxicity in APCs with the increase of melittin concentration.”

Page 8, line 208 of manuscript (Results: α -melittin-NPs have more cytotoxic effects on tumor cells than on APCs) now states the following: “In addition to zeta potential, cholesterol has been reported to have an inhibiting effect on the lytic activity of melittin to erythrocytes⁴⁸. Next, we estimated the cholesterol content in the three different cells using an Amplex® Red reagent-based assay. However, the data showed that the cholesterol level in B16F10 cells was significantly higher than that in BMDCs and BMDMs (**Supplementary Fig. 8**). These results suggest that membrane potential-mediated cellular binding and subsequent spatial distribution are, at least in part, the main reasons for differential killing effects in a certain concentration range.”

Supplementary Figure 6. Zeta potential measurement of B16F10 cells, BMDCs and BMDMs.

Data are shown as the mean \pm SEM. *** $P < 0.001$ and **** $P < 0.0001$, as analyzed by one-way ANOVA with Bonferroni's post hoc test.

Figure. 2d, e. Evaluation of cellular-binding ability of α -melittin-NPs. (d) Quantitative data of the MFI of FITC- α -melittin-NPs in B16F10 cells, BMDCs and BMDMs. Incubation time: 3 h. MFI: mean fluorescent intensity. The MFI values were normalized according to minimum (0 μ M) in each type of cell. (e) Representative immunofluorescence imaging of cellular binding of FITC- α -melittin-NPs (10 μ M) to B16F10 cells, BMDCs and BMDMs. Incubation time: 3 h. Blue: DAPI, green: FITC- α -melittin-NPs, red: membrane-targeted tdTomato. Scale bar: 5 μ m. Data are shown as the mean \pm SEM. * P <0.05, ** P <0.01 and **** P <0.0001, as analyzed by one-way ANOVA with Bonferroni's post hoc test.

Supplementary Figure 7. Quantitative data of the MFI of FITC- α -melittin-NPs in B16F10 cells, BMDCs and BMDMs. Incubation time: 1 h. MFI: mean fluorescent intensity. The MFI values were normalized according to minimum (0 μ M) in each type of cell. Data are shown as the mean \pm SEM. $**P < 0.01$, $***P < 0.001$ and $****P < 0.0001$, as analyzed by one-way ANOVA with Bonferroni's post hoc test.

Supplementary Figure 8. Estimation of cholesterol content in different cells. B16F10 cells, BMDMs and BMDCs were lysed in PBS containing 2% Triton X-100 for 10 min. After centrifugation (12000 rpm, 15 min), the resulting supernatant was used for detecting cholesterol content with an Amplex® Red reagent-based assay. Data are shown as the mean \pm SEM. $****P < 0.0001$, as analyzed by one-way ANOVA with Bonferroni's post hoc test.

In our previous study, we confirmed that low concentrations of α -melittin-NPs induced primarily early apoptosis in B16F10 cells and that high concentrations of α -melittin-NPs resulted primarily in B16F10 cell necrosis in vitro. As noted by the editor, we further investigated cell death in vivo to support the mechanism of action of α -melittin-NPs. In fact, the TUNEL assay was been applied to tumor tissue sections, not to cultured cells in our study (**Supplementary Fig. 5** in original version, **Supplementary Fig. 9** in revised version). tRFP-B16F10 tumor cells that expressed tetrameric far-red fluorescent protein (tRFP) were used to visualize the release of endogenous antigens as well as the change in integrity of the cell membrane. The fluorescence intensity of the TUNEL channel is very high even though the laser power (640 nm) is 1%. However, we needed to modify the method of image processing because we did not take into account the nonspecific binding of necrotic cells. By subtracting the fluorescence background in the TUNEL channel, the real TUNEL positive cells can be analyzed precisely. In addition, to make the contrast even more remarkable, we changed the pseudo-colour of the TUNEL channel from magenta to green (magenta might interfere with red). As shown in Fig. R3 below (or Fig. S9 above), the real TUNEL positive cells with nuclear structures (not diffuse fluorescence) were discernible, and only some B16F10 cells were TUNEL positive (apoptotic cells). It is noticeable, however, that TUNEL-positive cells also displayed the disappearance of fluorescent model antigens (plasma membrane permeabilization). It has been reported that apoptotic cells eventually lose plasma membrane integrity and progress to secondary necrosis in the absence of phagocytosis⁴⁹. Therefore, TUNEL-positive cells may have progressed to secondary necrosis characterized by plasma membrane permeabilization after the development of apoptotic DNA fragmentation.

In summary, although the small vesicles formed during apoptosis trap cellular contents as well as the fluorescent model antigens, the apoptotic cells induced by α -melittin-NPs may not have time to form small vesicles in vivo, instead experiencing secondary necrosis characterized by plasma membrane permeabilization.

Figure. R3. Representative immunofluorescence imaging of necrotic/apoptotic tumor cells induced by α -melittin-NPs. In the original version, the white asterisk represents the TUNEL positive cell, and the yellow asterisk represents the necrotic area. After subtracting the background fluorescence (nonspecific reaction because of necrotic cells) in the TUNEL channel, the real TUNEL positive cells with nuclear structure (not diffuse fluorescence) were discernible.

[redacted]

[redacted]

We modified the manuscript as follows:

Page 9, line 221 of manuscript (Results: α -melittin-NPs have more cytotoxic effects on tumor cells than on APCs) now states the following: “The fluorescence imaging of tumor tissue sections showed that both melittin and α -melittin-NPs induced the disappearance of fluorescent model antigens in the tumor section after 24 h, indicating the changes in membrane penetrability and the release of the tumor antigens (Supplementary Fig. 9). The immunofluorescence imaging of TUNEL staining indicated that TUNEL positive cells were also induced by melittin and α -melittin-NPs. It is noticeable, however, that TUNEL-positive cells also displayed the disappearance of fluorescent model antigens (plasma membrane permeabilization). Thus, α -melittin-NPs can shield the cytotoxicity of melittin to APCs, but maintain the ability of melittin to kill B16F10 tumor cells and induce the release of whole tumor cell antigens *in vivo*.”

3. The major data in this manuscript is a series of bilateral flank tumor experiments that are intended to verify “the whole cell vaccine” importance to melittins action on cancer cells. The results in this manuscript are striking and strongly support a role for melittin nanoparticles in cancer therapy. The contralateral tumor specific and prolonged suppression of tumor growth that the authors present are strong indications that this protocol has merit. The authors then address the role of cellular immunity verses serum antibody response. The cellular response was robust and specific. The IGG response seemed much less dramatic in that the melittin activated response was only twice that of the controls. Then the authors investigate the tumor environment resulting from melittin promoted lymphocyte infiltration. These experiments were presented as a single time point making it difficult to assess the chronological importance of the melittin induced changes.

Response:

We thank the reviewer for their constructive suggestion. According to the reviewer's suggestion, we performed these experiments at different time points (14 days and 21 days), and we added new data in **Fig. 6b-d** and **Supplementary Fig. 13** in the revised manuscript.

Cellular and humoral immune response:

As shown in **Supplementary Fig. 13**, α -melittin-NPs had no effect on the frequencies of cytokine⁺ (TNF- α ⁺ and IFN- γ ⁺) T cells 14 days after left tumor implantation. In addition, there was also no difference in the percentage of IgG⁺ cells between all the groups. The data at 21 days are shown in the original manuscript (Fig. 5). These results indicate that specific cellular immunity and humoral immunity did not occur in the early stages. Many studies also suggest that the time points to evaluate antigen-specific T cell responses and antibody responses should be selected at 21 days (or longer) after immunization or tumor implantation⁵⁶⁻⁵⁸.

We modified the manuscript as follows:

Page 10, line 289 of manuscript (Results: α -melittin-NPs elicit tumor antigen-specific cellular and humoral immune response) now states the following: "We used flow cytometry to analyze the cytokine expression in T cells. The data showed that **compared with PBS group**, α -melittin-NPs induced increases in the frequencies of IFN- γ ⁺CD8⁺ (12.2-fold) and IFN- γ ⁺CD4⁺ (7.2-fold) T cells **at day 21 after tumor implantations but no differences emerged at 14 days.** (**Fig. 5a-c and Supplementary Fig. 13a, b**)."

Lymphocyte infiltration:

Furthermore, we also analyzed lymphocyte infiltration in distant tumors at different time points (14 days and 21 days).

We modified the manuscript as follows:

Page 11, line 311 of manuscript (Results: α -melittin-NPs induce lymphocyte infiltration and dramatic changes in the cytokine/chemokine milieu in the distant tumor) now states the following: "**The flow cytometry data showed that, compared to the PBS, α -melittin-NPs induced an increase in the numbers of innate immune**

components, including natural killer (NK) cells, monocytes and neutrophils, but not the adaptive components, including CD4⁺ and CD8⁺ T cells at 14 days after left tumor implantation (**Fig. 6b, c**). However, at 21 days, the α -melittin-NP group exhibited a significant increase in the number of CD4⁺ (4.4-fold, $p = 0.0074$) and CD8⁺ T cells (3.7-fold, $p = 0.00243$) in addition to NK cells and monocytes. Immunofluorescence analysis of the distant tumors also revealed that CD4⁺ and CD8⁺ T cells were present at high density after α -melittin-NP treatment (**Fig. 6d**).”

These results suggest that inflammatory infiltration induced by α -melittin-NPs was shifting from mainly innate immune cells in the early stage to mainly adaptive immune cells in the late stage.

Supplementary Figure. 13 Antigen-specific T cells and antibody responses at day 14. (a, b) C57BL/6 mice ($n = 4$ per group) were treated as described above (Fig. 3a). On day 14, the lymphocytes isolated from the tumor-draining LNs were restimulated with DCs pulsed with B16F10 tumor lysates and were analyzed by flow cytometry with intracellular cytokine staining. Cytokine⁺ cell frequencies from each group are shown. (c) B16F10 tumor cells were incubated with 5% serum that was collected from treated mice and age-matched naïve mice. Subsequently, these cells were stained with a DyLight649-conjugated mouse IgG-specific secondary antibody and analyzed by flow cytometry. IgG⁺ cell frequencies from each group are shown. Error bars

indicate the SEM. Statistical analysis was performed with one-way ANOVA by Bonferroni's post hoc test.

Fig. 6 α -melittin-NPs induce lymphocyte infiltration. (a) Treatment scheme. (b, c) Absolute numbers of adaptive immune cells (b) and innate immune cells (c) in the distant tumors were calculated by flow cytometry. (d) Representative immunofluorescence images from distant tumors 21 days after left tumor implantation. Scale bar, 20 μ m. Data are shown as the mean \pm SEM (n = 3). n.s. not significant, * P < 0.05, ** P < 0.01 and *** P < 0.001, as analyzed by one-way ANOVA with Bonferroni's post hoc test.

Minor comments :

1. In the Discussion the authors refer to work by Dezfuli but I did not find a reference to this work in the manuscript.

Response:

We thank the reviewer for their constructive suggestion. In the revised manuscript, we provided the exact author's name (Page 13, line 358).

Reference

1. He, C. *et al.* Core-shell nanoscale coordination polymers combine chemotherapy and photodynamic therapy to potentiate checkpoint blockade cancer immunotherapy. *Nat. Commun.* **7**, 12499 (2016).
2. Twyman-Saint Victor, C. *et al.* Radiation and dual checkpoint blockade activate non-redundant immune mechanisms in cancer. *Nature* **520**, 373-377 (2015).
3. Mi, Y. *et al.* A dual immunotherapy nanoparticle improves T-cell activation and cancer immunotherapy. *Adv. Mater.* **30**, 1706098 (2018).
4. Min, Y. *et al.* Antigen-capturing nanoparticles improve the abscopal effect and cancer immunotherapy. *Nat. Nanotechnol.* **12**, 877-882 (2017).
5. Zamarin, D. *et al.* Localized oncolytic virotherapy overcomes systemic tumor resistance to immune checkpoint blockade immunotherapy. *Sci. Transl. Med.* **6**, 226ra32-226ra32 (2014).
6. Zamarin, D. *et al.* Intratumoral modulation of the inducible co-stimulator ICOS by recombinant oncolytic virus promotes systemic anti-tumour immunity. *Nat. Commun.* **8**, 14340 (2017).
7. Hamilton, J. R., Vijayakumar, G., & Palese, P. A recombinant antibody-expressing influenza virus delays tumor growth in a mouse model. *Cell Rep.* **22**, 1-7 (2018).
8. Nam, J. *et al.* Chemo-photothermal therapy combination elicits anti-tumor immunity against advanced metastatic cancer. *Nat. Commun.* **9**, 1074 (2018).
9. Kim, M. S. *et al.* Role of hypothalamic Foxo1 in the regulation of food intake and energy homeostasis. *Nat. Neurosci.* **9**, 901 (2006).
10. Chen, C. *et al.* Polyvalent nucleic Acid/Mesoporous silica nanoparticle conjugates: dual stimuli - responsive vehicles for intracellular drug delivery. *Angew. Chem. Int. Ed. Engl.* **50**, 882-886 (2011).
11. Soman, N. R. *et al.* Molecularly targeted nanocarriers deliver the cytolytic peptide melittin specifically to tumor cells in mice, reducing tumor growth. *J. Clin. Invest.* **119**, 2830-2842 (2009).

12. Lee, G. Y. et al. Hyaluronic acid nanoparticles for active targeting atherosclerosis. *Biomaterials* **53**, 341-348 (2015).
13. Jun, Y. W. et al. Addressing the autofluorescence issue in deep tissue imaging by two-photon microscopy: the significance of far-red emitting dyes. *Chem. Sci.* **8**, 7696-7704 (2017).
14. Hansch, A. et al. Autofluorescence spectroscopy in whole organs with a mobile detector system1. *Acad. Radiol.* **11**, 1229-1236 (2004).
15. Buehler, B. A. Vitamin B2: riboflavin. *Journal of Evidence-Based Complementary & Alternative Medicine*, **16**, 88-90 (2011).
16. Junttila, M. R., & de Sauvage, F. J. Influence of tumour micro-environment heterogeneity on therapeutic response. *Nature* **501**, 346-354 (2013).
17. Liu, Z., Ravindranathan, R., Kalinski, P., Guo, Z. S., & Bartlett, D. L. Rational combination of oncolytic vaccinia virus and PD-L1 blockade works synergistically to enhance therapeutic efficacy. *Nat. Commun.* **8**, 14754 (2017).
18. Liu, Z. et al. Modifying the cancer-immune set point using vaccinia virus expressing re-designed interleukin-2. *Nat. Commun.* **9**, 4682 (2018).
19. Pulido, J. et al. Using virally expressed melanoma cDNA libraries to identify tumor-associated antigens that cure melanoma. *Nat. Biotechnol.* **3**, 337 (2012).
20. Slingluff Jr, C. L. et al. Randomized multicenter trial of the effects of melanoma-associated helper peptides and cyclophosphamide on the immunogenicity of a multipeptide melanoma vaccine. *J. Clin. Oncol.* **29**, 2924-2932 (2011).
21. Ott, P. A. et al. An immunogenic personal neoantigen vaccine for patients with melanoma. *Nature* **547**, 217-221 (2017).
22. Sahin, U. et al. Personalized RNA mutanome vaccines mobilize poly-specific therapeutic immunity against cancer. *Nature* **547**, 222-226 (2017).
23. Schumacher, T. N., & Schreiber, R. D. Neoantigens in cancer immunotherapy. *Science* **348**, 69-74 (2015).
24. Sahin, U., & Türeci, Ö. Personalized vaccines for cancer immunotherapy. *Science* **359**, 1355-1360 (2018).

25. Wang, S. et al. Intratumoral injection of a CpG oligonucleotide reverts resistance to PD-1 blockade by expanding multifunctional CD8⁺ T cells. *Proc. Natl. Acad. Sci. USA*. **113**, E7240-E7249 (2016).
26. Singh, M. et al. Intratumoral CD40 activation and checkpoint blockade induces T cell-mediated eradication of melanoma in the brain. *Nat. Commun.* **8**, 1447 (2017).
27. Ribas, A. et al. Oncolytic virotherapy promotes intratumoral T cell infiltration and improves anti-PD-1 immunotherapy. *Cell* **174**, 1031-1032 (2018).
28. Sagiv-Barfi, I. et al. Eradication of spontaneous malignancy by local immunotherapy. *Sci. Transl. Med.* **10**, eaan4488 (2018).
29. Henninot, A., Collins, J. C., & Nuss, J. M. The current state of peptide drug discovery: back to the future? *J. Med. Chem.* **61**, 1382-1414 (2017).
30. Paull, B. R., Yunginger, J. W., & Gleich, G. J. Melittin: an allergen of honeybee venom. *J. Allergy Clin. Immunol.* **59**, 334-338 (1977).
31. Schiener, M., Graessel, A., Ollert, M., Schmidt-Weber, C. B., & Blank, S. Allergen-specific immunotherapy of Hymenoptera venom allergy—also a matter of diagnosis. *Hum. Vaccin. Immunother.* **13**, 2467-2481 (2017).
32. Spillner, E., Blank, S., & Jakob, T. Hymenoptera allergens: from venom to “venome”. *Front. Immunol.* **5**, 77 (2014).
33. Palm, N. W. et al. Bee venom phospholipase A2 induces a primary type 2 response that is dependent on the receptor ST2 and confers protective immunity. *Immunity* **39**, 976-985 (2013).
34. Sturm, G. J. et al. EAACI guidelines on allergen immunotherapy: Hymenoptera venom allergy. *Allergy* **73**, 744-764 (2018).
35. Yang, L. et al. Abrogation of TGFβ signaling in mammary carcinomas recruits Gr-1⁺ CD11b⁺ myeloid cells that promote metastasis. *Cancer cell* **13**, 23-35 (2008).
36. Chaneton, B., & Gottlieb, E. PGAMgnam style: a glycolytic switch controls biosynthesis. *Cancer cell* **22**, 565-566 (2012).
37. Arif, S. et al. Anti-TNFα therapy—killing two birds with one stone? *The Lancet* **375**, 2278 (2010).

38. Colonna, M. TREM1 Blockade: Killing Two Birds with One Stone. *Trends Immunol.* **40**(9), 781-783 (2019).
39. Hewing, B. et al. Effects of native and myeloperoxidase-modified apolipoprotein aI on reverse cholesterol transport and atherosclerosis in mice. *Arterioscler. Thromb. Vasc. Biol.* **34**, 779-789 (2014).
40. Kingwell, B. A. et al. HDL-targeted therapies: progress, failures and future. *Nat. Rev. Drug. Discov.* **13**, 445-464 (2014).
41. Silva, R. G. D. et al. Structure of apolipoprotein AI in spherical high density lipoproteins of different sizes. *Proc. Natl. Acad. Sci. USA* **105**, 12176-12181 (2008).
42. Huang, et al. Hybrid melittin cytolytic peptide-driven ultrasmall lipid nanoparticles block melanoma growth in vivo. *ACS nano* **7**, 5791-5800 (2013).
43. Phillips, M. C. New insights into the determination of HDL structure by apolipoproteins Thematic Review Series: High Density Lipoprotein Structure, Function, and Metabolism. *J. lipid Res.* **54**, 2034-2048 (2013).
44. T. Utsugi, et al. Elevated expression of phosphatidylserine in the outer membrane leaflet of human tumor cells and recognition by activated human blood monocytes. *Cancer research* **51**, 3062-3066 (1991).
45. Gajski, G., & Garaj-Vrhovac, V. Melittin: a lytic peptide with anticancer properties. *Environ. Toxicol. Pharmacol.* **36**, 697-705 (2013).
46. Melo, M. N., Ferre, R., & Castanho, M. A. Antimicrobial peptides: linking partition, activity and high membrane-bound concentrations. *Nat. Rev. Microbiol.* **7**, 245 (2009).
47. Yu, X. et al. Immune modulation of liver sinusoidal endothelial cells by melittin nanoparticles suppresses liver metastasis. *Nat. Commun.* **10**, 574 (2019).
48. Raghuraman, H., & Chattopadhyay, A. Cholesterol inhibits the lytic activity of melittin in erythrocytes. *Chem. Phys. Lipids* **134**, 183-189 (2005).
49. Poon, I. K., Lucas, C. D., Rossi, A. G., & Ravichandran, K. S. Apoptotic cell clearance: basic biology and therapeutic potential. *Nat. Rev. Immunol.* **14**, 166-180 (2014).

50. Napoletano, F. et al. p53-dependent programmed necrosis controls germ cell homeostasis during spermatogenesis. *PLoS Genet.* **13**, e1007024 (2017).
51. Nakasone, E. S. et al. Imaging tumor-stroma interactions during chemotherapy reveals contributions of the microenvironment to resistance. *Cancer cell* **21**, 488-503 (2012).
52. Kroemer, G. et al. Classification of cell death: recommendations of the Nomenclature Committee on Cell Death 2009. *Cell Death Differ.* **16**, 3(2009).
53. Berghe, T. V. et al. Determination of apoptotic and necrotic cell death in vitro and in vivo. *Methods* **61**, 117-129 (2013).
54. Kepp, O., Galluzzi, L., Lipinski, M., Yuan, J., & Kroemer, G. Cell death assays for drug discovery. *Nat. Rev. Drug Discov.* **10**, 221 (2011).
55. Cabral, H. et al. Accumulation of sub-100 nm polymeric micelles in poorly permeable tumours depends on size. *Nat. Nanotechnol.* **6**, 815 (2011).
56. Carroll, E. C. et al. The vaccine adjuvant chitosan promotes cellular immunity via DNA sensor cGAS-STING-dependent induction of type I interferons. *Immunity* **44**, 597-608 (2016).
57. Cao, Y. et al. Ultrasmall graphene oxide supported gold nanoparticles as adjuvants improve humoral and cellular immunity in mice. *Adv. Funct. Mater.* **24**, 6963-6971 (2014).
58. Moynihan, K. D. et al. Eradication of large established tumors in mice by combination immunotherapy that engages innate and adaptive immune responses. *Nat. Med.* **22**, 1402 (2016).

REVIEWERS' COMMENTS:

Reviewer #2 (Remarks to the Author):

The authors have thoroughly addressed the questions, and I have no further comments.

Reviewer #3 (Remarks to the Author):

In my initial review I made clear that I thought this manuscript was well executed and important but made several suggestions that I suggested would increase the conclusions available to the manuscript. The revised manuscript retains the strengths of the original manuscript. The authors have documented the preparation and characterized the particles more completely. The authors feel that FITC-a-mellittin-NPs cannot be directly applied to the cultured cell assays. However they feel that the characterization of the NPs is so similar that they must be similar in activity. While I do not ascribe to this correlation as completely as the authors I feel that they have made a good faith effort to answer my request and that their assertions for the effects of the NPs are reasonable. Therefore I feel they have addressed my first comment adequately. I will consider the authors response to Major comments 2 and 3 together as authors response considered some of the same issues in their point-by-point response. In my original critique I expressed concern that there were experiments the authors had not undertaken that prevented them from making mechanistic conclusions. In their response they have made clear that my suggestions were not possible in all cases but more than that they provided many of the experimental results I suggested. This has resulted in the adding of new data in Fig. 2d, e and Supplementary Fig. 6-8. The use of TUNEL to demonstrate apoptosis in tissues and the authors in depth analysis of these data in the revised version have enabled further interpretation which I feel adds to the significance of the manuscript. Therefore I would like to support the publication of this manuscript.